# Residue 6.43 defines receptor function in class F GPCRs

Ainoleena Turku [1,2], Hannes Schihada[1,3], Pawel Kozielewicz [1,3], Carl-Fredrik Bowin [1] & Gunnar Schulte [1✉]

The class Frizzled of G protein-coupled receptors (GPCRs), consisting of ten Frizzled (FZD$_{1-10}$) subtypes and Smoothened (SMO), remains one of the most enigmatic GPCR families. While SMO relies on cholesterol binding to the 7TM core of the receptor to activate downstream signaling, underlying details of receptor activation remain obscure for FZDs. Here, we aimed to investigate the activation mechanisms of class F receptors utilizing a computational biology approach and mutational analysis of receptor function in combination with ligand binding and downstream signaling assays in living cells. Our results indicate that FZDs differ substantially from SMO in receptor activation-associated conformational changes. SMO manifests a preference for a straight TM6 in both ligand binding and functional readouts. Similar to the majority of GPCRs, FZDs present with a kinked TM6 upon activation owing to the presence of residue P$^{6.43}$. Functional comparison of FZD and FZD P$^{6.43}$F mutants in different assay formats monitoring ligand binding, G protein activation, DVL2 recruitment and TOPflash activity, however, underlines further the functional diversity among FZDs and not only between FZDs and SMO.

[1] Karolinska Institutet, Department of Physiology & Pharmacology, Sec. Receptor Biology & Signaling, Biomedicum 6D, Stockholm, Sweden. [2] Present address: Orion Pharma R&D, Espoo, Finland. [3] These authors contributed equally: Hannes Schihada, Pawel Kozielewicz. ✉email: gunnar.schulte@ki.se

Human, nonsensory G protein-coupled receptors (GPCRs) are classified into the four main classes A, B, C, and F based on their sequence homology[1]. While high-resolution crystal or CryoEM structures are available for representatives of all these GPCR classes in the inactive state, the active, G protein-coupling or nanobody-stabilized, high-resolution structures are more rare—currently these have been resolved for different receptors of class A and class B, and one representative of class F[2]. The very first active GPCR structures were of rhodopsin at low pH[3] or bound to a peptide derived from the C-terminus of Gα subunit[4] in the year 2008, followed by a nanobody-stabilized β2 adrenoceptor and a β2 adrenoceptor-Gs complex from year 2011[5,6]. Studying GPCR activation has led to the common agreement that an outward movement of the transmembrane helix 6 (TM6) is the main hallmark of the active GPCR conformation[7,8]. This reorganization of the TM6 involves a complex interaction network within the receptor structures and is also tied to the presence of the conserved proline residues $P^{5.50}$, $P^{6.50}$, and $P^{7.50}$ allowing helix dynamics (for Ballesteros-Weinstein numbering, see refs. [9,10]). These proline residues are highly conserved within class A GPCRs, but missing for some part in the other GPCR classes[10].

The class F or class Frizzled, which consist of ten Frizzled (FZD1–10) subtypes and Smoothened (SMO) remains one of the most enigmatic groups of GPCRs[11,12]. Class F receptors are intrinsically involved in embryonic development, tissue homeostasis and pathology, most prominently oncogenesis[13,14], thus emphasizing the need for a better understanding of the molecular mechanisms of receptor activation. FZDs mediate WNT signaling and SMO mediates hedgehog signaling, but underlying mechanisms of ligand binding, receptor activation and signal specification and initiation remain obscure[15]. Our recent work on class F structure-function aspects emphasized that two distinct, conformational activation pathways exist in FZDs. On the one hand, interaction with G proteins is accompanied by the opening of a molecular switch ($R^{6.32}/W^{7.55}$) in the receptor core between TM6 and TM7. On the other hand, mutational disruption of the molecular switch prevents interaction with DVL at the same time as it enhances WNT-5A-induced recruitment of miniG proteins[16] indicating that FZDs employ different active conformations for pathway selectivity.

While inactive SMO and FZD structures have provided detailed insight into the overall receptor structure, the first high-resolution structures of active class F receptors were reported in 2019 showing active SMO in complex with either heterotrimeric Gi protein or a nanobody, and currently six active SMO structures are available[15,17–19]. Keeping the preconception of the bent TM6 causal for GPCR activation in mind, it was unexpected that the outward movement of the TM6 in these SMO structures originated from a movement of the whole helix rather than merely bending it (Fig. 1a)[17,18]. Even though SMO and FZDs bear structural similarities in both transmembrane ligand and intracellular effector binding sites[16,20] and interact with and activate heterotrimeric G proteins (for full references see refs. [12,15]), SMO signaling is different from FZD signaling. The differences are most distinct when comparing ligand binding and downstream signaling along the FZD/β-catenin or the SMO/GLI pathways[21,22]. Based on what is known from class A and B GPCRs and SMO regarding the conformational changes occurring upon receptor activation, we employ here computational biology and mutational analysis of receptor function to shed light on details of class F receptor activation mechanisms.

Several lines of evidence point into the direction of ligand-induced and constitutive activity of class F receptors resembling the typical mode of action of class A and B GPCRs including an opening of the TM bundle by a TM6 swing out, which can for example be monitored using conformational FZD biosensors[20,23,24] and through the recruitment of miniG proteins as conformational sensors of the active conformation[16]. While FZDs all have a proline at position 6.43 ($P^{6.43}$) in TM6, SMO carries a phenylalanine underlining differences in receptor activation mechanisms and cholesterol binding in these closely related GPCRs. We propose that SMO activation relies on cholesterol binding to the 7TM core of the receptor maintaining a straight TM6, whereas activation-associated conformational changes of FZD6 rather manifest in a kinked TM6 and are independent of cholesterol binding in the 7TM core of the receptor. A better understanding of differences in class F receptor activation offers possibilities to develop small molecule compounds targeting both SMO and FZDs with higher selectivity and improved pharmacological profile.

## Results

### MD simulations of SMO and FZD6 model reveal different TM6 topologies.

In order to broadly compare all class F GPCRs, we performed a sequence alignment of class F receptors in human, orangutan, rat, mouse, dog, chicken, frog, and fruit fly based on the sequences obtained from the UniProtKB/Swiss-Prot database (Supplementary Data 1). In the evolutionarily conserved class F, only $P^{5.50}$ is conserved similarly as reported for class A; location 6.50 is occupied by a cysteine residue and 7.50 by an isoleucine residue (Supplementary Data 1). While TM7 lacks proline residues almost throughout the class F receptors (only in FZD3 amino acid 7.51 is a proline), TM6 harbours a $P^{6.43}$ in all FZDs but not in SMO (Fig. 1b). This is also in line with the outcome of a previously published large scale sequence alignment of over 750 mammalian and non-mammalian FZDs and SMO (Supplementary Fig. 1) underlining the strict conservation of the $P/F^{6.43}$ across the animal kingdom[16].

This striking difference between FZDs and SMO regarding position 6.43 raised the question how the presence of a $P^{6.43}$, which is unable to complete the hydrogen bonding network of the α-helix and thus interrupts the preferred helical geometry in TM6, affects activation-associated conformational rearrangements in FZDs compared to SMO. In order to assess, whether $P^{6.43}$ indeed affects the overall conformation of active FZDs, we utilized a homology model of FZD6 built using active SMO (PDB ID: 6OT0) as a template[20]. The sequence of SMO is most similar to that of FZD3 and FZD6[16]. Of these, we selected FZD6 for the MD simulations due to the fact that simulating FZD6 allowed us to utilize the SMO agonist SAG1.3 in maintaining the simulated receptors in an active-like conformation in absence of the intracellular effector similar to what we have reported before[20]. The best representative model, as well as the active SMO, were then used as a starting structure for MD simulations.

In these MD simulations, a distinct kink was observed in the TM6 of FZD6 (angle 158.5° ± 4.5° throughout the MD trajectory run in four independent replicas comprising 1250 ns of simulation per simulation system), whereas the TM6 remained notably straighter in SMO (168.4° ± 4.2°; Fig. 1c–e). To further evaluate, whether this difference in angle was indeed mediated only by $P^{6.43}$, we set up similar MD simulations with mutated FZD6 ($P^{6.43}F$) and SMO ($F^{6.43}P$). The MD data with the point-mutated receptors inevitably indicated that the observed bend of FZD6 was mediated by $P^{6.43}$: the TM6 of $P^{6.43}F$ FZD6 remained in the straight conformation, whereas the TM6 of $F^{6.43}P$ SMO was bending (Fig. 1d–e). For further details of these MD simulations, see Supplementary Fig. 2 (protein backbone RMSDs) and Supplementary Fig. 3 ($\chi_2$ torsion angles of $P^{6.43}$ to monitor the ring puckering states).

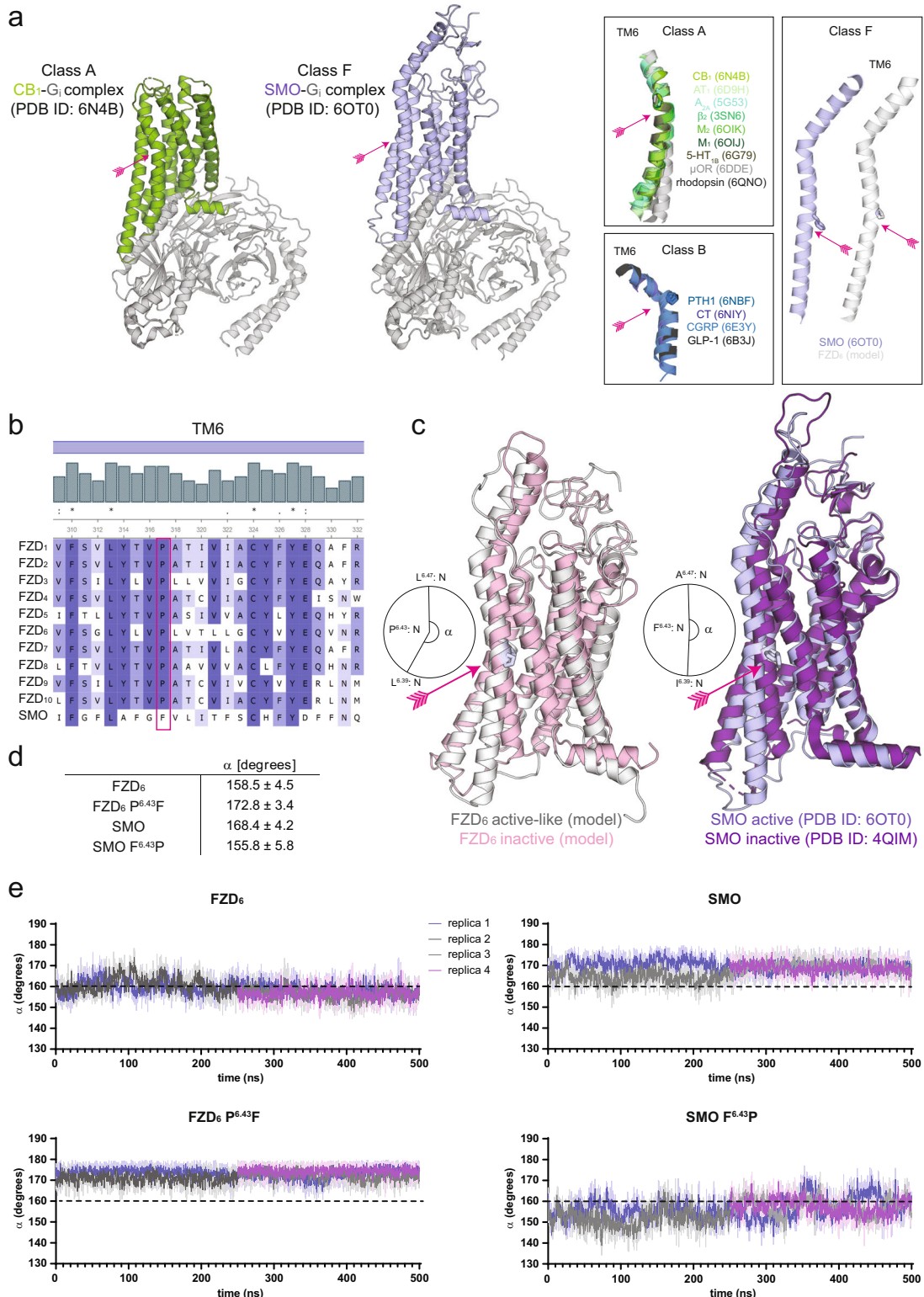

**Fig. 1 The overall conformations of TM6s in active GPCRs. a** $CB_1$ and SMO example structures (left), conformations of TM6s of class A, B, and F GPCRs (right). $Pro^{6.50}$, $Pro^{6.47}$, and $Pro/Phe^{6.43}$, respectively, are shown as sticks, while receptors and G proteins are shown as cartoon (class A: green, class B: blue, SMO: violet, $FZD_6$: white, $G_i$: gray). In each structure, the kink in the TM6 is marked with a pink arrow. **b** Sequence alignment of human class F receptors (see also Supplementary Data 1). The conserved $Pro^{6.43}$ in FZDs is marked with a pink rectangle. The grey bars and violet shadings indicate the sequence conservation. The horizontal violet bar indicates the location of TM6. **c** Active and inactive structures of $FZD_6$ (left) and SMO (right). Active receptors are shown as white ($FZD_6$) and violet cartoon (SMO), and incative receptors as pink ($FZD_6$) and dark violet (SMO) cartoon. **d**, **e** TM6 angles throughout the MD trajectory. The angle is measured between the backbone nitrogen atoms of $Leu/Ile^{6.39}$, $Pro/Phe^{6.43}$, and $Leu/Ala^{6.47}$ in each MD frame. In **d** angles are averaged and given as mean ± SD, whereas in **e** each measured angle is given as a function of time. The simulation replicas 1 (blue) and 2 (dark gray) started from $t = 0$ ns are plotted starting from time point 0 and replicas 3 (light gray) and 4 (violet) started from $t = 250$ ns are plotted starting from time point 250. Thick traces indicate the moving average smoothed over a 1 ns window and thin traces the raw data.

To control further that the observed kink in TM6 is not caused by the lack of the intracellular effector, we set up a control simulation with wild-type FZD$_6$-SAG1.3-miniG$_i$ protein complex (ca. 550 ns of simulation divided into three replicas; Supplementary Fig. 4). The utilized miniG$_i$ protein construct acts as a conformational sensor for active FZD$_6$[16,20], and we selected it for these control simulations instead of the full G$_i$ due to its notable smaller size. In these control simulations, the TM6 orientation remained similar to the active-like model of the wild-type FZD$_6$ without miniG$_i$ (Supplementary Fig. 4).

**F$^{6.43}$P (in SMO) mutation affects ligand binding and 7TM pocket volume.** SMO crystal structures, both active and inactive, have shown that the transmembrane binding site for small molecular ligands extends deeply inside the 7TM core of the receptor[17,25]. Furthermore, our recent study implementing a NanoBRET binding assay for SMO, suggested the existence of two binding pockets especially in the ΔCRD-SMO (SMO lacking the extracellular cysteine-rich domain) assessed by BODIPY-cyclopamine binding[26]. In presence of the extracellular CRD, BODIPY-cyclopamine binds mainly to the high-affinity (lower) binding site of SMO, which is also targeted by cholesterol and synthetic antagonists of the SANT series[17,25]. For FZD$_6$ instead, our previous BODIPY-cyclopamine binding data support only binding to the pocket that is analogous to the low-affinity (upper) binding site of SMO that is also targeted by the SAG series of SMO agonists[20].

During class A GPCR activation, the outward-swing of TM6 is accompanied by reorganization of several amino-acid residues (including so-called microswitches) throughout the receptor structure[27]. Some of these structural rearrangements take also place at the orthosteric ligand binding cavity underlining the allosteric cooperativity between G protein binding and the extracellular ligand binding site as originally defined in the ternary complex model[28]. With the intramolecular interaction networks and the deep binding cavity of SMO in mind, we assessed whether also the FZD$_6$ and SMO cavities are affected by the conformational changes in TM6. In order to quantify the consequence of the alterations in the ligand binding cavity, we employed a NanoBiT/BRET-based binding assay using BODIPY-cyclopamine and HiBiT-tagged receptors[15,20,29] (Fig. 2a and b). For these experiments, we used wild-type HiBiT-SMO and HiBiT-FZD$_6$, together with their respective mutants: F$^{6.43}$P and P$^{6.43}$F. BODIPY-cyclopamine binding to HiBiT-SMO resulted in a monophasic and saturable concentration-dependent bioluminescence resonance energy transfer (BRET) signal (pK$_d$ ± SD = 6.87 ± 0.13). Most importantly, we observed a dramatic decrease (P = 0.0105) in the BODIPY-cyclopamine binding affinity to HiBiT-SMO F$^{6.43}$P (pK$_d$ ± SD = 5.44 ± 1.02). Interestingly, there were no differences between BODIPY-cyclopamine association to HiBiT-FZD$_6$ and HiBIT-FZD$_6$ P$^{6.43}$F (pK$_d$ ± SD = 6.45 ± 0.23 vs. 6.28 ± 0.20, P = 0.5945). In order to confirm that the reduction of BODIPY-cyclopamine in HiBiT-SMO F$^{6.43}$P monitors the lower binding site in SMO, we also compared wild-type and F$^{6.43}$P mutant ΔCRD-SMO, which exposes also the upper binding site as previously shown[26]. In support of our assumption, the binding data with BODIPY-cyclopamine argue that the upper binding site is accessible to BODIPY-cyclopamine in ΔCRD HiBit-SMO wild-type and F$^{6.43}$P (Supplementary Fig. 5). Of note, the NanoBiT/BRET-based binding assay monitors only the signal originated from the receptors expressed on the cell surface, as LgBiT is not cell-permeable.

As demonstrated by the decreased BODIPY-cyclopamine binding when comparing wild-type SMO to the SMO F$^{6.43}$P, the lower binding pocket of SMO seems to be disrupted by the F$^{6.43}$P mutation. Furthermore, the unaffected BODIPY-cyclopamine binding in FZD$_6$ P$^{6.43}$F compared to the wild-type receptor indicated that straightening out TM6 in FZD$_6$ neither affected the upper binding site nor enabled the lower binding site to bind BODIPY-cyclopamine. In order to obtain deeper insights into this, we monitored the binding pocket volumes of SMO (wild-type and F$^{6.43}$P) and FZD$_6$ (wild-type and P$^{6.43}$F) throughout the MD trajectories (Fig. 2c and d). In both wild-type receptors, the binding cavity volume was higher than that of the corresponding mutant. In case of the mutant SMO F$^{6.43}$P, the bent TM6 clearly reduced the volume of the lower pocket, whereas the upper pocket even slightly increased in volume (Fig. 2c; right panels). In FZD$_6$, the upper pocket remained similar despite the point mutation, but also here the volume below it was reduced; in FZD$_6$ the reduction appeared to be driven by a tight inter- and intrahelical interaction network involving TMs 3, 6, and 7 (Fig. 2c, left panels; see also the interaction network chapter below).

The effect of the F$^{6.43}$P mutation on the observed BODIPY-cyclopamine binding to SMO could thus be explained by the notable reduction in the volume of the lower binding cavity. Similarly, the almost similar BODIPY-cyclopamine binding to FZD$_6$ wild-type and P$^{6.43}$F mutant can be explained by the relatively minor changes at the upper binding cavity. Notably, the shapes and volumes of the 7TM binding cavities of the wild-type FZD$_6$ and SMO are different—even though they both can accommodate SAG1.3[20]. Also, apart from the upper pocket, the 7TM cavities of FZD$_6$ and SMO are in slightly different locations; in FZD$_6$ the additional cavity is lined by TMs 1, 2, 3, and 7, whereas it is close to TMs 3, 5 and 6 in SMO (Fig. 2c). However, in the FZD$_6$-miniG$_i$ control simulations, the additional volume close to TM2 was missing and the TM2 conformation resembled that of FZD$_6$ P$^{6.43}$F (Fig. 2c and Supplementary Fig. 4); this suggests that the additional space at the lower part of the 7TM pocket of wild-type FZD$_6$ (run without the miniG$_i$) could be a simulation artefact. Altogether, these observations underline that the lower ligand binding pocket does not exist in FZD$_6$ as it does in SMO and explains why BODIPY-cyclopamine interacts only with the upper 7TM pocket in FZD$_6$[20].

**Aromatic π-π interaction network extends the molecular switch.** We have previously identified a conserved molecular switch between R$^{6.32}$ (K$^{6.32}$ in FZD$_4$ and FZD$_9$) and W$^{7.55}$, which stabilizes the inactive state in class F receptors (Supplementary Fig. 7)[16]. Upon activation, the hydrogen bond between the positively charged side chain of R$^{6.32}$ and the backbone oxygen atom of W$^{7.55}$ breaks, presumably allowing the outward movement of TM6. When studying the binding cavities of the active-like FZD$_6$ and SMO here, we observed that there is an extended network of aromatic π interactions between the previously described molecular switch and P/F$^{6.43}$ in both FZD$_6$ and SMO (Fig. 2c, Supplementary Fig. 4, Supplementary Fig. 8 and Supplementary Fig. 9). In FZD$_6$, the network includes residues Y$^{6.40}$, W$^{3.43}$, F$^{6.36}$, and W$^{7.55}$, while in SMO there are A$^{6.40}$ and F$^{3.43}$ instead, rendering the network less extensive. The shorter network in SMO leaves space within the 7TM core of the receptor close to TM5 and TM6 forming the lower ligand (cholesterol) binding pocket[17,19,26].

Apart from the MD data, the aromatic interaction network is present in the crystal/cryoEM structures of active SMO (PDB IDs: 6OT0, 6O3C, 6XBJ, 6XBK, 6XBL, and 6XBM; Supplementary Fig. 7); however the molecular switch is open only in three structures (6OT0, 6O3C, and 6XBL). In the structures with a closed molecular switch, helix 8 is not resolved, rendering the precise location of the switch W$^{7.55}$ less certain. In the inactive SMO structures (PDB IDs 4QIM, 4O9R, 4N4W, 4JKV, and 5L7I), the molecular switch is closed and F$^{6.36}$ is facing towards W$^{7.55}$

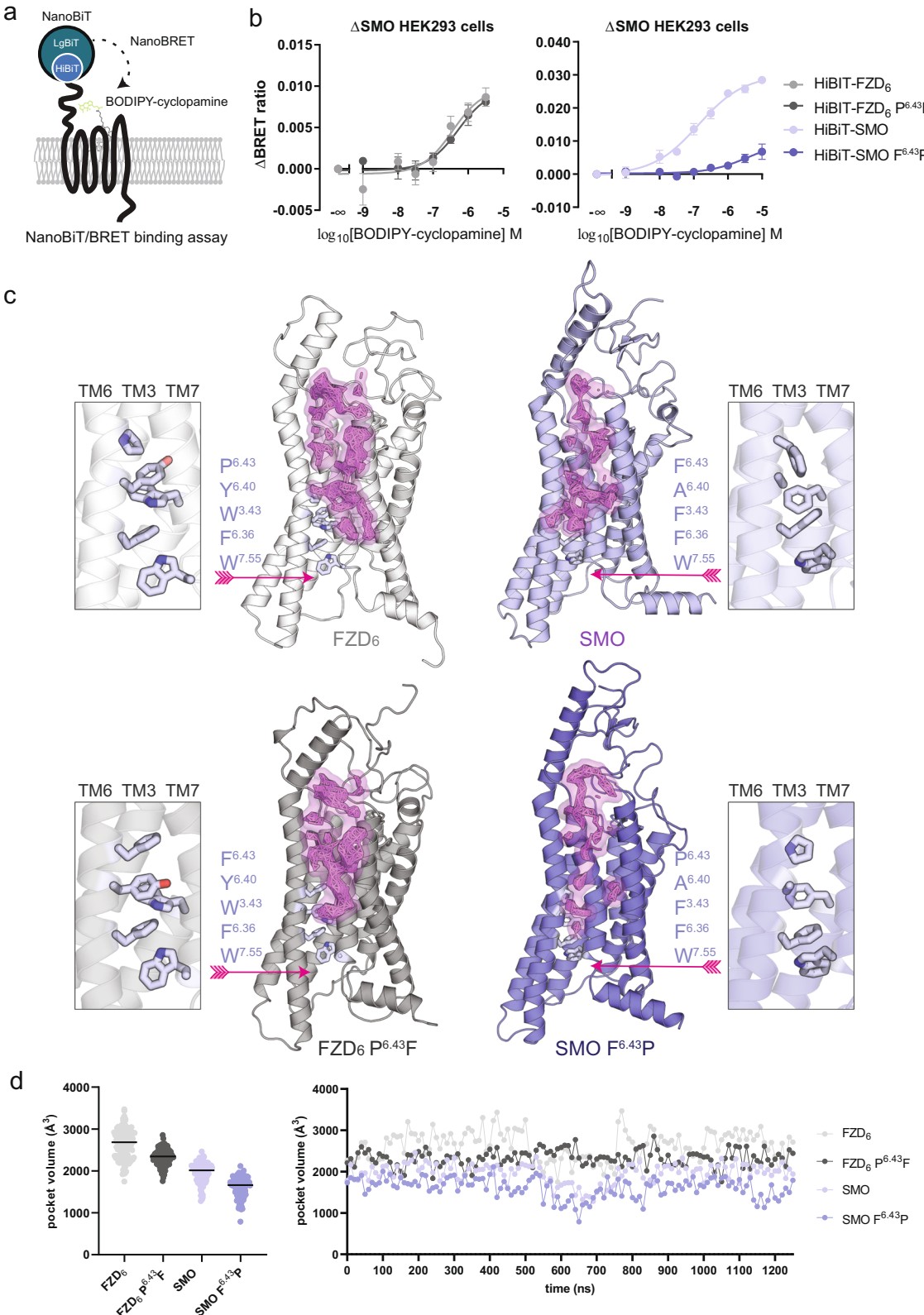

breaking the contact with $F^{3.43}$. The only FZD structures currently available are $FZD_4$ and $FZD_5$ (both in the ligand-free apo form; PDB IDs: 6BD4 and 6WW2, respectively;[30,31]), where the same interactions are observed—$F^{6.36}$ is facing towards $W^{7.55}$ and is not interacting with $W^{3.43}$. Additionally, $Y^{6.40}$ and $Y^{2.51}$ in the apo $FZD_4$ and $FZD_5$ structures are within a hydrogen bonding distance ($\leq 4$ Å). In SMO, a similar hydrogen bond is not possible, as there are $A^{6.40}$ and $F^{2.51}$ instead. In order to extrapolate these findings also to $FZD_6$, we built an inactive $FZD_6$ model based on SMO and $FZD_4$ crystal structures. In the inactive

**Fig. 2 BODIPY-cyclopamine binding and binding cavity volumes of FZD$_6$ and SMO. a** Assay design schematic. **b** BODIPY-cyclopamine binding to FZD$_6$ and FZD$_6$ P$^{6.43}$F (light and dark gray, respectively; left panel) and SMO and SMO F$^{6.43}$P (light and dark violet, respectively; right panel) was assessed by the NanoBiT/BRET-binding assay. Saturation curves are presented as sigmoidal curves with logarithmic BODIPY-cyclopamine concentrations. Graphs present net NanoBRET values. Data points are presented as mean ± SEM from n = 4 for HiBiT-FZD$_6$ P$^{6.43}$F and n = 5 individual experiments for the other receptor constructs. Curves for HiBiT-FZD$_6$ were fit to a three parameter model. For HiBiT-SMO curves were fit according to a four-parameter model. See Supplementary Fig. 6a–d for the cell-surface expression data of the HiBiT-receptor constructs. Source data are provided as a Source Data file. **c** The 7TM binding cavity of FZD$_6$ and SMO wild-type (upper panels) and P$^{6.43}$F and F$^{6.43}$P mutants (lower panels). The pink grid represents the locations of the Voronoi vertices at isovalue 3 (i.e., the location of the followed cavity in the protein structure) throughout the MD simulation trajectories. The receptors (the last frame of each system after 500 ns of simulation) are shown as cartoon and the aromatic interaction network (marked by pink arrows) as sticks. Color code is as follows: white cartoon: FZD$_6$, gray cartoon: FZD$_6$ P$^{6.43}$F, light violet cartoon: SMO, dark violet cartoon: SMO F$^{6.43}$P, violet sticks: carbon, red sticks: oxygen, blue sticks: nitrogen. Note that for visualization purposes the binding cavity grid is projected on top of the protein structures. **d** The binding cavity volumes of each studied system (left). The median is marked as black line. The cavity volumes as function of time (right). The replicas are plotted as a continuous trajectory (replica 1: 0–500 ns; replica 2: 500–750 ns; replica 3: 750–1000 ns; replica 4: 1000–1250 ns). The color code follows that of panel **c**.

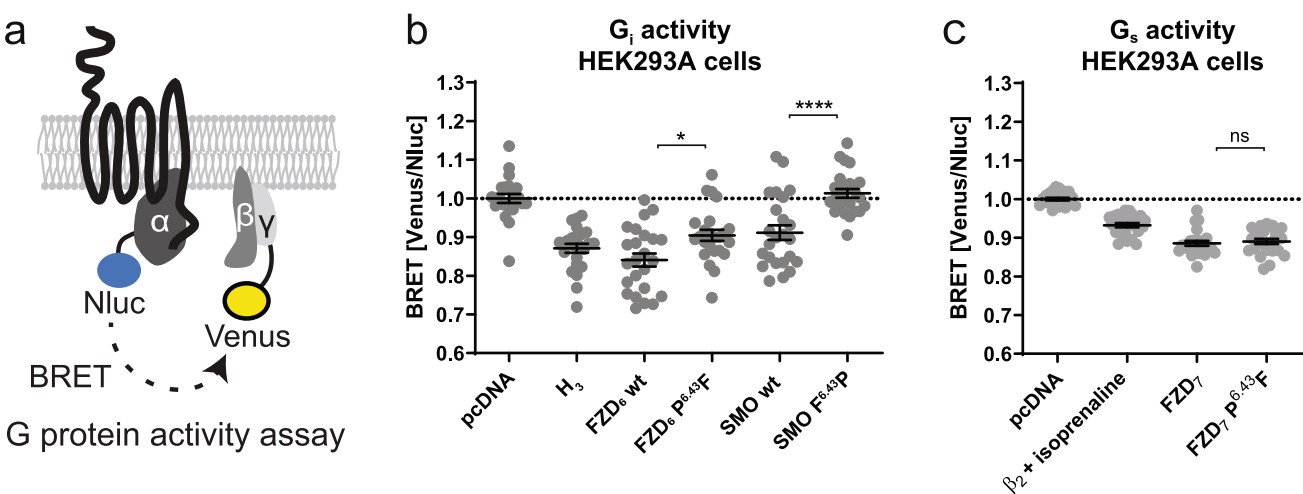

**Fig. 3 G protein activation by FZDs and SMO. a** Schematic presentation of G$_{i1/s}$ sensor design and assay principle. **b** G$_{i1}$ activation by FZD$_6$ and SMO wild-type and mutants. **c** G$_s$ activation by FZD$_7$. Empty vector (pcDNA) transfection was employed as a negative control and the histamine H$_3$ receptor (**b**) and β$_2$AR stimulated with 10 µM isoprenaline (**c**) were used as positive controls. Data are represented as mean ± SEM of three independent experiments measured in octuplicates. Statistical difference was assessed using One-Way ANOVA followed by Sidak's multiple comparison. *P < 0.05; ****P < 0.0001; ns not significant. All experiments were conducted in HEK293A cells transiently co-transfected with the indicated G protein sensor and receptor. See Supplementary Fig. 6e for the cell-surface expression of the utilized FZD$_6$, FZD$_7$, and SMO constructs. Source data are provided as a Source Data file.

FZD$_6$ model, the Y$^{6.40}$–Y$^{2.51}$ hydrogen bond is observed, while the aromatic network between W$^{3.43}$ and F$^{6.36}$ is broken. Notably, Y$^{2.51}$ and Y$^{6.40}$ are conserved throughout all ten FZDs but not in SMO (Supplementary Fig. 7, Supplementary Data 1).

**SMO F$^{6.43}$P mutation abolishes constitutive G$_i$ protein activity.** Based on the indications that the F$^{6.43}$P replacement in SMO (and P$^{6.43}$F in FZD$_6$) correlates with reduced volume of the lower 7TM cavity, we aimed to test experimentally whether such a change in receptor topology has any functional consequences. Both SMO and FZD$_6$ interact with common heterotrimeric G proteins and act as GPCRs leading to the activation of pertussis-sensitive G$_{i/o}$ signalling[16,18,20,27,32–35]. Therefore, we examined the effect of the SMO F$^{6.43}$P and FZD$_6$ P$^{6.43}$F mutations on the constitutive activity of SMO and FZD$_6$ towards heterotrimeric G$_i$ proteins using a BRET-based G$_{i1}$ sensor (Fig. 3a and b, for cell-surface expression of the receptor constructs, see Supplementary Fig. 6e). Additionally, we assessed constitutive activity of G$_s$-coupled FZD$_7$ with a BRET-based G$_s$ sensor (Fig. 3c)[16,20,36,37]. The contribution of endogenous, autocrine acting WNTs to receptor activation was excluded by pharmacological inhibition of porcupine using 10 nM C59 preincubation overnight.

As observed previously, wild-type SMO exerts significant constitutive activity in HEK293 cells as a consequence of low

endogenous levels of its inhibitor PATCH1[35,38–41]. In order to relate the constitutive activity of FZD$_6$ and SMO towards the G$_{i1}$ BRET sensor to that of a bona fide constitutively active GPCRs from class A, we included the histamine H$_3$ receptor in the panel[42]. Interestingly, the levels of constitutive G$_{i1}$ protein input of H$_3$ receptor, FZD$_6$ and SMO are in a similar range (not normalized for receptor surface expression). The dramatic abolishment of the activity towards G$_{i1}$ of SMO upon F$^{6.43}$P mutation suggests that the lower 7TM ligand binding site of SMO is required for maintaining this activity of SMO towards the G$_i$ pathway. This conclusion is also in line with the active SMO structures[17,19], where the lower 7TM cavity is occupied by a cholesterol when the receptor is in the active conformation. The constitutive activity of FZD$_6$, however, is not as much affected by the P$^{6.43}$F mutation as SMO activity is by F$^{6.43}$P mutation; this indicates that the lower 7TM pocket of FZD$_6$ does not play a role in FZD$_6$-mediated G$_i$ activation. However, it seems that the bent TM6 of FZD$_6$ is slightly preferable over the straight TM6 (as seen in FZD$_6$ P$^{6.43}$F mutant) in maintaining the constitutive activity of these receptors towards G$_i$. The constitutive activity towards heterotrimeric G$_s$ of FZD$_7$ that is in the same range and even stronger than the activity of isoprenaline-stimulated adrenergic β$_2$ receptors (see also ref. [37]), is not affected by the P$^{6.43}$F mutation. Notably, the cell-surface expression of the SNAP-tagged FZD$_6$ is

not affected by the point mutation indicating that the effect observed here is due to the change in the ability of $FZD_6$ to activate $G_{i1}$ upon point mutation and not the inability of the mutant to efficiently be presented at the cell surface (Supplementary Fig. 6e). Even though SMO $F^{6.43}P$ is slightly less expressed than wild-type SMO, the change in the $G_{i1}$ activity is clearly of different magnitude than that of the surface expression (Fig. 3b and Supplementary Fig. 6e). Furthermore, the trend in SMO $F^{6.43}P$ surface expression is similar to that of $FZD_7$ $P^{6.43}F$, whereas the G protein activity signals of these SNAP-tagged receptors are notably different. This supports further that the dramatic effect of the $F^{6.43}P$ mutation on the $G_{i1}$ activity of SMO is due to the direct change in the receptor conformation and not cell-surface expression.

In order to compare the effect of the $F^{6.43}P$ and $P^{6.43}F$ mutations in SMO and $FZD_6$, respectively, on agonist-induced receptor activation, we assessed the ability of SAG1.3 to elicit miniG$_i$ protein (mGsi) recruitment in accordance with previous findings[16,20]. Since miniG proteins serve as conformational sensors of the active state of GPCRs, this readout monitors both ligand binding and conformational changes in the receptor. In accordance with the data on the constitutive activity of these receptors, the mutations of residue 6.43 extinguished the SAG1.3-induced mGsi recruitment to SMO, while that to $FZD_6$ remained

unaffected (Supplementary Fig. 10). Note that the SAG1.3-concentration response curves follow the bell shape, which is characteristic for this ligand[20,43].

**SMO $F^{6.43}P$ represents a distinct receptor conformation from the active-state SMO.** SMO mediates GLI signaling in a G protein-independent manner relying on sequestration of the catalytic subunits of cyclic AMP-dependent protein kinase (cPKA) to the phosphorylated C-terminus of SMO[44]. In order to assess the consequences of the previously described topological changes of SMO for this additional signaling pathway, we measured recruitment of cPKA (fluorescently tagged cPKA-YFP) to wild-type and $F^{6.43}P$ SMO (with C-terminal Nluc tag) in direct BRET titration experiments (Fig. 4a). Intriguingly, SMO $F^{6.43}P$ was still able to interact with the cPKA, though to a notably lesser extent than wild-type SMO (Fig. 4b).

Since the SMO $F^{6.43}P$ mutant was still able to recruit cPKA albeit to a lesser degree, we set up to monitor whether the $F^{6.43}P$ mutation changes the SMO conformation as dramatically as our MD data suggested. As a direct readout of SMO conformation in living cells, we quantified the recruitment of a recently described conformational sensor, the YFP-tagged nanobody NbSmo2 (NbSmo2-YFP; Fig. 4c), which exclusively interacts with the active-state SMO and competes with NbSmo8 that was used for

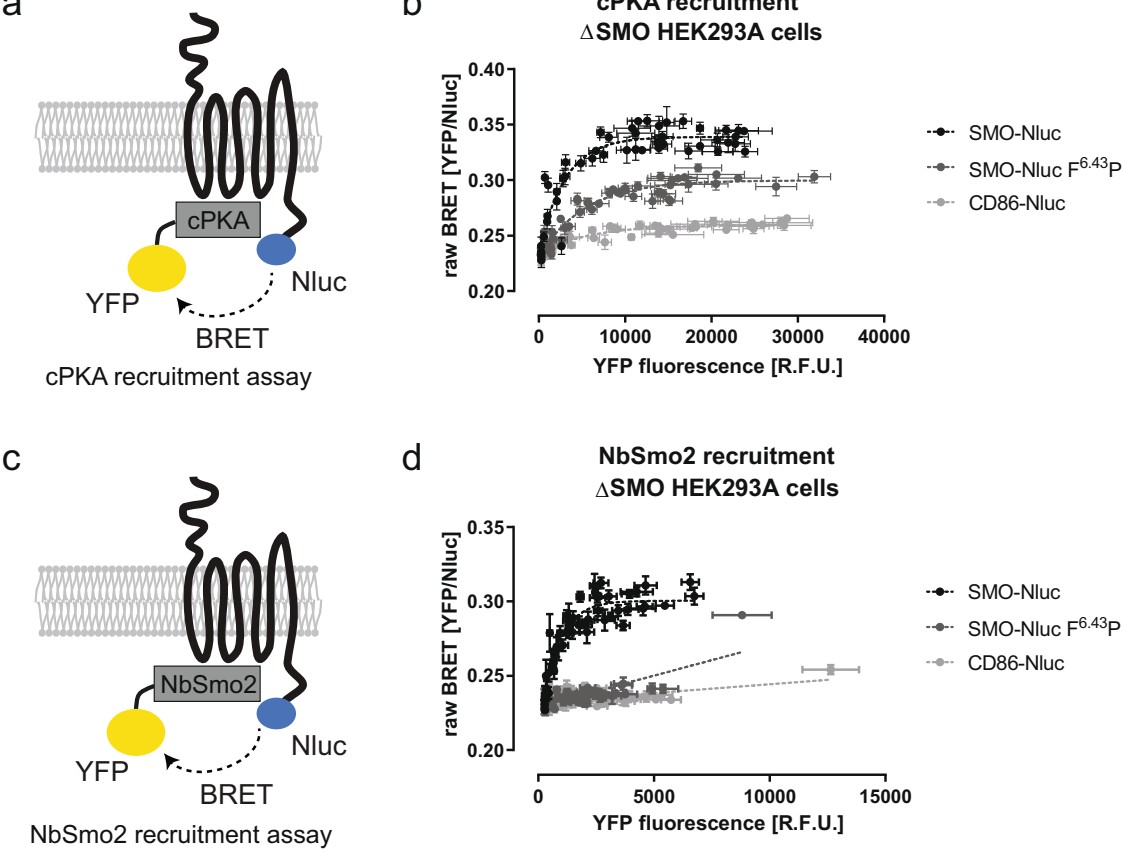

**Fig. 4 cPKA and NbSmo2 recruitment to SMO. a** Schematic presentation of the cPKA recruitment assay measuring direct BRET. **b** cPKA-YFP recruitment to SMO wild-type (black) and mutant (dark gray). CD86 (light gray) was employed as a negative control. **c** Scheme of the NbSmo2 recruitment assay measuring direct BRET. **d** NbSmo2-YFP recruitment to SMO wild-type (black) and mutant (dark gray). CD86 (light gray) was employed as a negative control. Experiments shown in **b** and **d** were conducted in ΔSMO HEK293 cells transiently co-transfected with the indicated receptor-Nluc construct and varying amounts of cPKA-YFP (**b**) or NbSmo2-YFP (**d**), respectively. Data show mean ± SD from four individual experiments conducted in quadruplicates. Each dataset was fitted to a one-phase association or linear model and the preferred fit was selected based on an Extra sum-of-squares F-test ($P < 0.05$). See Supplementary Fig. 6f for the cell-surface expression data of the receptor-Nluc constructs. Source data are provided as a Source Data file.

stabilizing the active conformation of SMO for CryoEM[17,44]. While wild-type SMO recruited NbSmo2, the SMO $F^{6.43}P$-NbSmo2-YFP BRET titration curve resembled a linear fit, similar to that of the negative control CD86-Nluc (Fig. 4d). This indicates that the $F^{6.43}P$ mutation in SMO indeed stabilizes a receptor conformation that is substantially different from that of the active-state SMO, and thus incapable to interact with the active-state specific nanobody. Notably, this finding also suggests that the BRET signal observed in the cPKA recruitment assay with the $F^{6.43}P$ SMO reflects either a specific receptor conformation (i.e., different from that of the active-state SMO detected by NbSmo2-YFP binding) to recruit cPKA or that cPKA interaction depends only on phosphorylation sites on intracellular epitopes of the receptor, which are not, or only slightly, affected by structural rearrangements resulting from the $F^{6.43}P$ mutation.

**$P^{6.43}F$ mutation interferes with DVL-dependent mechanisms.** Unlike SMO, FZDs interact with the phosphoprotein dishevelled (DVL) acting at the crossroads of WNT/β-catenin and WNT/planar cell polarity signaling[45]. Thus, assessing the interaction of DVL with FZDs provides yet another readout to relate FZD conformation with signal initiation. In WNT/β-catenin signaling, the recruitment of DVL1-3 to for example $FZD_4$, $FZD_5$ and $FZD_7$ reduces degradation of the transcriptional regulator β-catenin, ultimately leading to its nuclear translocation and regulation of so-called WNT target genes[14]. Of note, the β-catenin signaling cascade initiated by DVL recruitment to FZDs functionally resembles the GLI signaling cascade initiated by the cPKA recruitment to SMO[44]. In addition, $FZD_6$, which is predominantly associated with β-catenin-independent WNT/PCP-like signaling, also recruits DVL[16,46]. Thus, we assessed the effect of altered receptor topology on DVL2 recruitment to the plasma membrane by $FZD_4$, $FZD_5$, $FZD_6$, and $FZD_7$ and their corresponding $P^{6.43}F$ mutants using a bystander BRET assay (Fig. 5a). DVL2 recruitment to $FZD_5$ and $FZD_6$ was significantly reduced by the $P^{6.43}F$ mutation suggesting that the bent TM6 conformation plays a role in the receptor-DVL-complex formation (Fig. 5b). The DVL2 recruitment to $FZD_7$ and $FZD_4$ and their corresponding $P^{6.43}F$ mutants, however, were similar (Fig. 5b, Supplementary Fig. 6e and Supplementary Fig. 11). As DVL2 recruitment to the membrane is correlated with the receptor surface expression of the recruiting FZD, the presented data are normalized to the surface expression of the individual FZD constructs as shown in Supplementary Fig. 6e.

Since $FZD_6$ showed the most pronounced difference in the recruitment of full-length DVL2 comparing wild-type and the $P^{6.43}F$ receptors, we investigated this receptor paralogue further with an additional assay employing the DVL2 DEP domain as a conformational sensor. The DEP domain interacts with FZDs and in particular with $FZD_6$[46,47]. Here, we established a bystander BRET assay assessing Nluc-DEP recruitment to Venus-KRas in dependence of SNAP-$FZD_6$ surface expression (Fig. 5c). Since DEP recruitment is directly proportional to receptor construct surface expression, we plotted the bystander BRET over the SNAP-surface signal originating from the SNAP-$FZD_6$ or SNAP-$FZD_6$ $P^{6.43}F$ constructs labelled with membrane impermeable SNAP substrate. The bystander BRET over a broad receptor construct surface expression range emphasized that $FZD_6$ $P^{6.43}F$ presents with altered ability to recruit Nluc-DEP indicating that the overall conformation of $FZD_6$ $P^{6.43}F$ does not accommodate the DEP domain the same way as the wild-type receptor. Assessing SMO function using DVL recruitment assays is not suitable because SMO does not recruit DVL2 (Supplementary Fig. 11).

In order to assess how mutation of residue 6.43 affects agonist-induced and receptor-mediated signaling, we employed the TCF/

LEF reporter TOPflash assay reporting WNT/β-catenin signaling in $\Delta FZD_{1-10}$ HEK293 cells[48]. Agonist stimulation of $FZD_4$, $FZD_5$, and $FZD_7$ but not $FZD_6$ or SMO-transfected cells (WNT-3A for FZDs or SAG1.3 for SMO) resulted in an increased TOPflash signal (Fig. 5d and e), which is in agreement with the understanding of class F receptor pathway selectivity[49]. Mutation $P^{6.43}F$ reduced the ability of $FZD_4$ and $FZD_5$ to evoke β-catenin-dependent signaling in response to WNT-3A, whereas $FZD_7$-mediated effects were not affected. The decrease needs to be seen in the light of the surface expression data for the receptor constructs though (Supplementary Fig. 6e), since $FZD_5$—but not $FZD_4$—$P^{6.43}F$ shows lower surface expression compared to the wild-type. As our previous findings indicate that WNT/β-catenin signaling evoked by WNT-3A in HEK293 cells is independent of heterotrimeric G proteins[50], these findings most likely reflect the ability of these receptors to communicate through DVL. Indeed, this statement is supported by the bystander BRET data with $FZD_6$ (Fig. 5c) pinpointing the changed DVL DEP recruitment in $FZD_6$ $P^{6.43}F$ compared to $FZD_6$.

## Discussion

Our data provide a structural and functional distinction between SMO and the ten FZDs clarifying what was previously surmised, namely that SMO activation depends on interaction with a cholesterol molecule accommodated by a straight TM6. FZDs on the other hand act independently of cholesterol binding to the 7TM core, and FZD activation—depending on the pathway and FZD paralogue—involves a proline-based kink in TM6. These findings provide therefore deeper insight into the molecular details of the activation of SMO and most importantly of FZDs of which no active, effector-bound high-resolution structures exist so far.

While our data presented here suggest that FZDs and SMO are activated differently, it should be noted that also other GPCRs, which are structurally even more distinct, are able to activate heterotrimeric G proteins. Most notably, the degree to which TM6 kinks differs dramatically, for example when comparing class A to class B GPCRs (Fig. 1a), underlining that an enormous structural variability in the receptor topology can accommodate heterotrimeric G proteins and even the same G protein subtypes. Thus, it is not as controversial as it would first seem that also FZDs and SMO could be adopting different conformations while interacting with and activating the same G proteins. Even though the atomistic details of the active $FZD_6$ and SMO conformations are quite different (for more detailed discussion see below), the existence of the previously reported molecular switch mechanism between TM6 and TM7—and the extension of it as we present here—emphasizes that class F receptors follow some superfamily-wide concepts of receptor activation despite their structural differences. However, our findings underline as well that the paralogues of the FZD family are different, not only in their pathway selectivity or their selectivity to heterotrimeric G proteins but also in their dependence on the $P^{6.43}$-mediated kink.

In most of the class A GPCRs, the inactive receptor conformation is stabilized by the allosteric $Na^+$ ion coordinated by $D^{2.50}$ complexing amino-acid residues from TMs 1, 2, 3, 6, and 7[51]. Unlike other class A GPCRs, the actions of visual rhodopsin do not involve $Na^+$, but there its function in stabilizing the inactive receptor structure is replaced by a hydrogen bond interaction between $W^{6.48}$ and $Y^{7.48}$[52]. Similarly, class F crystal structures do not reveal the presence of the allosteric $Na^+$ ion, and the amino acid at location 2.50 is a cysteine throughout the receptor class (Supplementary Data 1). This suggests that the allosteric $Na^+$ binding site does not exist in class F receptors.

As observed from the inactive $FZD_4$ and $FZD_5$ structures (PDB IDs: 6BD4 and 6WW2, respectively) and the inactive $FZD_6$ model

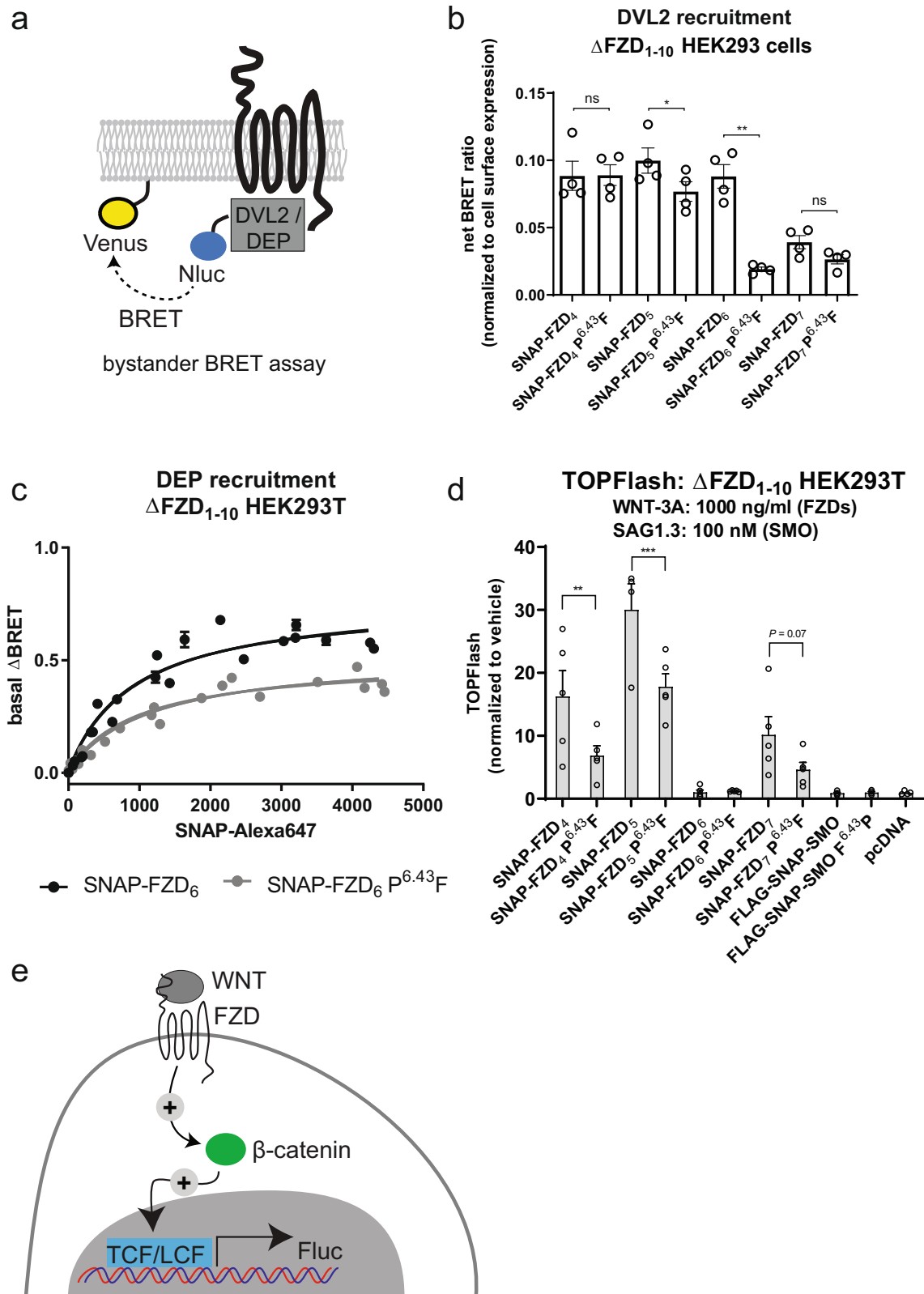

(Supplementary Fig. 7), $Y^{6.40}$ of $FZD_{4/5/6}$ is at hydrogen bonding distance from $Y^{2.51}$. These interaction counterparts are conserved throughout the FZDs (Supplementary Data 1), emphasizing an important role in the function of these receptors – possibly similar to the stabilizing effect of $W^{6.48}$–$Y^{7.48}$ on the inactive receptor conformation in visual rhodopsin. Interestingly, when

removing SAG1.3 from the $FZD_6$ wild-type simulation system, $Y^{6.40}$ starts to move slowly towards $Y^{2.51}$ (Supplementary Fig. 9); however, the 500 ns simulation time is naturally not enough for observing the full change towards the inactive receptor conformation. In SMO, an equivalent hydrogen bond is not possible due to the lack of both interaction counterparts ($F^{2.51}$ and $A^{6.40}$ in

**Fig. 5 Effects of $P^{6.43}F$ mutants in FZDs on DVL2 recruitment and WNT-induced signaling. a** Schematic presentation of the DVL2 bystander BRET assay. **b** DVL2 recruitment to $FZD_4$, $FZD_5$, $FZD_6$, and $FZD_7$. Bystander BRET ratio changes (between Nluc-DVL2 and Venus-KRas) were assessed in $\Delta FZD_{1-10}$ HEK293 cells in the presence of overexpressed wild-type and mutated SNAP-tagged $FZD_4$, $FZD_5$, $FZD_6$ and $FZD_7$. The data were normalized to the receptor surface expression levels and are shown as mean ± SEM of four independent experiments. Data were analyzed for each wild-type/mutant pair using paired two-tailed $t$-test. *$P < 0.05$, **$P < 0.01$, ns not significant. $P = 0.0146$ for $FZD_5$ wild-type vs. mutant and $P = 0.0027$ for $FZD_6$ wild-type vs. mutant. **c** Refined bystander BRET between the Nluc-tagged DEP domain of DVL2 and Venus-KRas, where the BRET values are plotted over the signal for the surface-expressed SNAP-$FZD_6$ (black) or SNAP-$FZD_6$ $P^{6.43}F$ (gray). Data show basal DEP recruitment in dependence of the surface-expressed receptor construct in the presence of C59. Data are shown as mean ± SD of four independent experiments. **d** WNT-3A (1000 ng/ml)-induced TOPflash signaling in $\Delta FZD_{1-10}$ HEK293 cells in the presence of overexpressed wild-type and mutated SNAP-tagged $FZD_4$, $FZD_5$, $FZD_6$, and $FZD_7$. TOPflash data are normalized to the vehicle control for each individual receptor isoform and mutant. Data were analysed using one-way ANOVA with Fisher's LSD test. **$P < 0.01$, ***$P < 0.001$. $P = 0.0030$ for $FZD_4$ wild-type vs. mutant and $P = 0.0004$ for $FZD_5$ wild-type vs. mutant. Data are presented as mean ± SEM of four independent experiments. **e** Schematic presentation of the TOPflash transcriptional reporter assay. See Supplementary Fig. 6e for the cell-surface expression data of the SNAP-tagged receptors. Source data are provided as a Source Data file.

SMO). Also, SMO is notably rich in small hydrophobic residues pointing towards the lower 7TM binding site and no apparent polar interactions seem to be present in that region of the inactive SMO crystal structures. This again is most likely one of the key factors in the capability of SMO to accommodate cholesterol at the lower 7TM pocket[17,19]. It is possible, that lack of cholesterol solely stabilizes inactive SMO while FZDs rely more on intramolecular interactions similar to visual rhodopsin.

The recently solved structures of SMO-cholesterol complexes (PDB IDs: 6O3C, 6XBJ, 6XBK, and 6XBL) show that the 7TM binding site of SMO accommodates cholesterol at the bottom (i.e., the lower pocket), middle and upper parts of the binding site[17,19]. Also, those data strongly suggest that the cholesterol molecule enters the lower 7TM pocket directly from the membrane from between the TMs 5 and 6. The simulation data that we present here agree with these findings, since the 7TM binding pocket extends outside the 7TM core through TMs 5 and 6 in SMO wild-type simulations (Supplementary Fig. 12). In $FZD_6$ instead, the binding pocket remains in the middle of the transmembrane helices at this receptor region. This difference seems to be due to the bulkier residues at the locations 5.59, 6.37, and 6.40 in $FZD_6$ when compared to SMO (L/G$^{5.59}$, S/G$^{6.37}$, and Y/A$^{6.40}$ in $FZD_6$ and SMO, respectively; Supplementary Fig. 12). This is also in line with the point mutation study of Deshpande et al.[17], where G$^{5.59}$F and A$^{6.40}$F point mutations rendered SMO partially or fully insensitive to stimulation.

As presented in Supplementary Fig. 8, the aromatic network remains stable in the majority of the simulations; however, some exceptions are observed. In replica 3 of $FZD_6$ wild-type simulation, W$^{3.43}$ withdraws from the network by flipping towards TM2 (Supplementary Fig. 8, upper-left panel). Subsequently, the remaining part of the network reorganizes bringing F$^{6.40}$ closer to F$^{6.36}$. Similar reorientation of the network amino acids is seen in replica 2 of SMO F$^{6.43}$P simulation (Supplementary Fig. 8). Another visible fluctuation is present in $FZD_6$ wild-type simulation replica 2, where W$^{7.55}$ moves further from F$^{6.36}$ (Supplementary Fig. 8). However, the range of the aromatic π–π interaction can be up to 7.5 Å and thus some attraction between the network counterparts is most likely maintained despite these changes. In general, it is hard to judge, whether these reorientations reflect reality, as the simulations are run in absence of the intracellular G protein; it is possible that some of the inconsistencies we observe are merely simulation artefacts. Notably, the aromatic network fluctuations discussed above remained absent in the $FZD_6$ -SAG1.3-miniG$_i$ simulations (Supplementary Fig. 4).

In conclusion, obtaining active FZD conformation involves different structural rearrangements compared to SMO. $FZD_6$ not only lacks the lower 7TM ligand binding pocket, which could accommodate cholesterol, but also the 'gate' between TMs 5 and 6, through which cholesterol could enter the receptor from the

inner leaflet of the lipid bilayer. Instead, $FZD_6$ (and FZDs in general) harbors a conserved proline residue in TM6, which allows the helix to bend similar to what is observed in other GPCR classes. The kinked conformation of TM6 is stabilized by the extensive aromatic network involving residues from TMs 3, 6, and 7, and the inactive $FZD_6$ conformation is stabilized in analogy to the intramolecular stabilization in visual rhodopsin. Generally, it appears that FZDs can accept the mutation of $P^{6.43}$ since receptor functionality is only partially impaired or not affected by the mutation. This could indicate that additional residues can contribute to TM6 dynamics such as the completely conserved G$^{6.34}$ aiding the receptor to present with an open conformation to accommodate e. g. the heterotrimeric G protein. Previous sequence alignment and analysis of class F receptor mutations occurring naturally in humans indicated that the natural variability of residue 6.43 is comparatively high-emphasizing that FZDs with a mutated $P^{6.43}$ are not primarily associated with disease[16]. However, SMO requires the straight TM6 conformation in G protein activation and prefers it for cPKA recruitment. All this is in line with the active SMO structures, which have shown that cholesterol plays a notable role in SMO activation; however, no active FZD structure is yet available. Our analysis predicts though, that future structural high-resolution insight through CyroEM or crystal structures of activated FZDs will reveal a TM6 kink emphasizing the role of $P^{6.43}$ in FZDs.

Even though the SMO data we present here are rather distinct—that is that removing the cholesterol binding site reduces the SMO activity in all tested readouts—the magnitude of the reduction is not identical in the different assays. Both G protein activation and NbSmo2 recruitment recognizing the active conformation of SMO (against the G protein pathway), are totally abolished by the F$^{6.43}$P mutation indicating that the lack of cholesterol keeps SMO F$^{6.43}$P mutant in a conformation, which is completely G$_i$ inactive (Fig. 3b and Fig. 4d). Interestingly, SMO F$^{6.43}$P mutant maintains also some of its ability to recruit cPKA (Fig. 4b), suggesting that the SMO conformation required for that interaction is also slightly different than the receptor conformation required when interacting with G proteins.

The interpretation of the data concerning FZDs is more complex, and unfortunately, no unifying FZD-family-wide conclusion is apparent. In case of $FZD_6$, the bent TM6 is preferred in G$_i$ activity, however, also the straight conformation is able to activate G$_i$ to some level (Fig. 3b). This might be because $FZD_6$ does not rely on the lower 7TM pocket and cholesterol binding in forming the active conformation, and thus the functional outcome of the changed TM6 topology is not as dramatic as in SMO (where the lack of the lower 7TM pocket means directly the lack of cholesterol, which in turn means the inaccessibility of the G$_i$-active receptor conformation). The G$_s$ activity of $FZD_7$ is not

 

affected by the $P^{6.43}F$ mutation suggesting even higher flexibility in the G protein-active conformation (Fig. 3c), however we cannot specify whether $G_s$ is more tolerant to the straight TM6 topology than $G_i$ or whether the overall $FZD_7$ topology is less affected by the point mutation than that of $FZD_6$.

In the DVL2 recruitment instead, $FZD_6$ $P^{6.43}F$ mutation strongly reduces the signal, and the trend is apparent also with $FZD_5$ (Fig. 5b, Supplementary Fig. 11). As $FZD_6$ is mainly associated with WNT/planar cell polarity-like signaling and not WNT/β-catenin signaling, it seems possible that the receptor conformation required for DVL2 recruitment to $FZD_6$ differs from that of the three other monitored FZDs. However, the TOPflash data (Fig. 5d) show that the ligand-induced WNT/β-catenin signaling mediated by $FZD_4$ and $FZD_5$ is also reduced by the $P^{6.43}F$ mutation indicating that the TM6 topology has a role in the active conformations of these FZD paralogues towards WNT/β-catenin signaling pathway as well. This could be due to the disturbance the $P^{6.43}F$ mutation causes to the complex formation with other involved proteins, such as LRP5/6.

It should be underlined that the presented assays comparing the ability of wild-type and mutant receptor to recruit transducers are affected by the cell-surface expression of the respective receptor construct. Thus, it is essential to take the effect of mutations on receptor trafficking into account when validating the receptor's constitutive activity. For this purpose, we provide surface expression data for all receptor constucts used in this study (Supplementary Fig. S6) and provide acceptor titration experiments, wherever possible and meaningful. Comparing the effects of the receptor mutants on cell-surface expression and function, however, we are confident that the observed functional alterations are indeed caused by the mutants directly and cannot be explained by differences in cell-surface expression.

Altogether, our study provides new insights onto the molecular details of class F activation mechanisms and highlights differences between SMO and FZDs. In the context of our recent findings that SMO-targeting small molecule ligands can be repurposed for FZDs[20], the appreciation of mechanistic details and differences among class F receptors indeed offers new opportunities for developing compounds that target these receptors with improved selectivity and pharmacological profile.

## Methods

**Homology modelling.** Twenty homology models of active-like $FZD_6$ (UniProt ID: O60353) were built with MODELLER 9.22[53] using active 24(S),25-epoxy-ycholesterol- and $G_i$-bound SMO structure (PDB ID: 6OT0[18]) as a template[20]. The sequence identity between the template (ΔCRD-SMO) and modelled ΔCRD-$FZD_6$ is approximately 29% and sequence similarity approximately 48%. To select a starting model for MD simulations, SAG1.3 was docked to these models with Glide software[54–56] in the Schrödinger Maestro 2018-4 molecular modeling platform— the model producing SAG1.3 docking pose best resembling the pose of SAG1.5 in complex with SMO (PDB ID: 4QIN[25]) was selected as a starting point for the further MD simulations.

For homology modelling of the inactive conformation of $FZD_6$ we used the inactive $FZD_4$ (PDB ID: 6BD4[30], receptor core and ICL areas) and SMO (PDB IDs: 4QIN[25] and 5L7D[57], ECLs and extended TM6 helix) crystal structures as templates[24]. The sequences of these proteins were aligned with ClustalX2[58] and 20 models were built with MODELLER 9.22. The model with the highest DOPE score was selected to be used in the structural comparisons.

$MiniG_i$ was modelled with a similar procedure as the inactive $FZD_6$ using the $miniG_s$ protein of PDB ID: 5G53 as a template. The most stabile $miniG_i$ construct, $miniG_{s/i1-43}$, by Nehmé et al.[59] was selected as the modelled $miniG_i$ sequence.

**Molecular dynamics (MD) simulations.** MD simulations were performed on the models of the active-like $FZD_6$, and the active SMO (PDB ID: 6OT0) using GROMACS 2019.2[60]. As we were interested in the active receptor conformations, the systems were "stimulated" by the presence of SAG1.3. To obtain the SMO-SAG1.3 starting complex for the simulation, the 24(S),25-epoxycholesterol was removed and SAG1.3 heavy atoms copied to the structure from the SAG21k-SMO complex (PDB ID: 6O3C;[17]). SAG1.3 was parametrized with AmberTools18 package using GAFF2 force field and AM1-BCC charges[61–63].

The $FZD_6$-SAG1.3-miniG_i complex was built using guidance from SMO-$G_i$ crystal structure (PDB ID: 6OT0) and $FZD_6$-$G_i$ docking poses (constructed with HADDOCK 2.2[64,65]). The full-length $G_i$ from the crystal structure was docked to $FZD_6$ active-like model (see above) with default parameters and the pose that resembled most the SMO-$G_i$ complex was selected as a reference complex. Then, the $miniG_i$ model was superimposed to the full-length $G_i$ of the reference complex, and the $FZD_6$ model was copied from the reference complex to the $miniG_i$ structure.

The receptors were oriented using the OPM database[66] and embedded in the POPC lipid bilayer (150 lipids/leaflet) by CHARMM-GUI server[67] with TIP3p water molecules and 0.15 M NaCl. The system was minimized for approximately 3000 steps and subsequently equilibrated with gradually decreasing position restraints on protein and lipid components. In the last 50 ns of the equilibration run, the harmonic force constants of 50 kJ mol$^{-1}$ nm$^{-2}$ were applied on the protein and ligand atoms only.

The $FZD_6$ $P^{6.43}F$ and SMO $F^{6.43}P$ mutants were constructed from the equilibrated structures by point mutation wizard of PyMol (The PyMOL Molecular Graphics System, Version 2.0 Schrödinger, LLC). After the point mutation, the receptors were placed back to the equilibrated systems and the receptor topology files were updated. Then, the systems were subjected to an additional energy minimization of ~1000 steps.

The independent isobaric and isothermic (NPT) ensemble production simulations for each system were initiated from random velocities using the CHARMM36m force field[68] and a 2 fs time step. The temperature at 310 K was maintained with Nose-Hoover thermostat[69] and the pressure at 1 bar with Parrinello-Rahman barostat[70]. Potential-shift-Verlet was used for electrostatic and van der Waals interactions with 12 Å cutoff. The bonds between hydrogen and other atoms were constrained by the LINCS algorithm[71]. Original receptor-SAG1.3 systems were simulated for 1250 ns divided into 4 independent replicas. Replicas 1 (500 ns) and 2 (250 ns) were started from frame $t = 0$ ns (i.e., directly after the equilibration), whereas replicas 3 and 4 (250 ns/each) were started from frame $t = 250$ ns of replica 2. $FZD_6$-SAG1.3-miniGi system was simulated as three independent replicas all started from $t = 0$ ns comprising ca. 550 ns of simulation in total.

The MD simulation data were generally analyzed using VMD 1.9.3 (visualization and measurement of RMSDs, distances and angles) and visualized in PyMol. Exception to this was the aromatic network distances, which were measured with the CPPTRAJ via AmberTools18[62]. The binding site volumes were monitored with MDpocket of the Fpocket suite[72,73]. For MDpocket, multi-PDB files including one frame/10 ns of each trajectory (126 poses/system in total) were used as an input. The grid points (at isovalue of 3) contributing to a continuous pocket inside the 7TM core were selected to be monitored in the pocket volume analysis. MD simulations will be made available at the open access server, GPCRmd: the MD database for GPCRs (www.gpcrmd.org). Snapshots of the receptor models extracted from the MD trajectories are provided as Supplementary Data 2–21.

**Cell culture and transient transfection.** Human embryonic kidney cells (HEK293A) wild-type (female origin; Thermo Fisher Scientific, R70507), ΔSMO HEK293A[20] or ΔFZD$_{1-10}$ HEK293T cells[74] were used for transient expression and grown in DMEM supplemented with 10% fetal calf serum, 0.1 mg/ml streptomycin, and 100 units/ml penicillin at 37 °C with 5% CO$_2$. All transfections were performed using Lipofectamine 2000 in a 2 μl Lipofectamine 2000/μg total DNA ratio. Absence of mycoplasma contamination was routinely confirmed by PCR using 5′-ggc gaa tgg gtg agt aac acg-3′ and 5′-cgg ata acg ctt gcg act atg-3′ primers detecting 16 S ribosomal RNA of mycoplasma in the media after 2–3 days of cell exposure.

**Cloning of receptor constructs and mutagenesis.** To generate HiBiT-SMO and ΔCRD HiBiT-SMO, Nluc sequence in Nluc-SMO or ΔCRD Nluc-SMO (coding mouse SMO, ref. [26]) were replaced with HiBiT sequence (nucleotides sequence: 5′-GTG AGC GGC TGG CGG CTG TTC AAG AAG ATT AGC-3′; amino acids sequence: VSGWRLFKKIS). To generate FLAG-SNAP-SMO, mouse SMO sequence from SMO-$Rluc8$ was inserted into an empty FLAG-SNAP-tagged pcDNA3.1 vector between BamHI and HindIII sites. SMO-$Rluc8$, HiBiT-$FZD_6$, SNAP-$FZD_4$, SNAP-$FZD_5$, SNAP-$FZD_6$, SNAP-$FZD_7$, $FZD_4$-Nluc, $FZD_6$-Nluc, β$_2$AR, Venus-KRas, and Nluc-DVL2 were generated and validated in our previous studies[16,29]. SMO-Nluc and the $G_s$ BRET sensor were generated using prolonged overlap extension PCR techniques. The plasmid encoding CD86-Nluc was provided by Dr. Ulrike Zabel (University of Wuerzburg, Wuerzburg, Germany). Plasmids encoding cPKA-YFP and NbSmo2-YFP were a kind gift from Benjamin Myers (University of Utah, Salt Lake City, USA)[44]. Plasmid encoding Venus-mGsi was from Nevin Lambert (Augusta University, Georgia, USA)[75]. Wild-type human H$_3$R DNA vector was purchased from cDNA.org. The desired mutations were generated using GeneArt site-directed mutagenesis kit (Thermo Fisher Scientific). All constructs were validated by sequencing (Eurofins GATC, Konstanz, Germany). Utilized primers are listed in Supplementary Table 1.

**NanoBiT/BRET-binding assay.** ΔSMO HEK293A cells[20] were transiently transfected in suspension using Lipofectamine®2000 (Thermo Fisher Scientific). $4 \times 10^5$

 

cells were transfected in 1 ml with 50–200 ng of HiBiT-tagged receptor plasmid DNA and 800–950 ng of pcDNA plasmid DNA. The cells (100 µl) were seeded onto a poly-D-lysine-coated black 96-well cell culture plate with solid flat bottom (Greiner Bio-One). Twenty-four hours post-transfection, the cells were washed once with 200 µl of HBSS (HyClone). Next, the cells were incubated with different concentrations of BODIPY-cyclopamine (80 µl) in HBSS for 90 min at 37 °C with $CO_2$. Subsequently, 80 ul of mix of furimazine (1:100 dilution; Promega) and LgBiT (1:200 dilution; Promega) were added, and the cells were incubated for another 10 min prior to the BRET measurements. The BRET ratio was determined as the ratio of light emitted by BODIPY (energy acceptor) and light emitted by HiBit-tagged receptors (energy donors). The BRET acceptor (bandpass filter 535–30 nm) and BRET donor (bandpass filter 475–30 nm) emission signals were measured using a CLARIOstar microplate reader (BMG). BODIPY fluorescence was measured prior to reading BRET (excitation: 470–15 nm, emission: 515–20 nm). The data were presented as net BRET (average raw BRET ratio of all BODIPY-cyclopamine unlabelled control wells were subtracted from the raw BRET ratios of all BODIPY-cyclopamine labelled wells). Cell-surface expression of HiBiT-tagged receptors was assessed by measuring luminescence of vehicle-treated wells (no BRET acceptor) in the NanoBiT/BRET-binding assay. The data were analyzed using GraphPad Prism 6.

**BRET-based measurement of $G_{i/s}$ activity**. HEK293A cells were transfected in suspension with plasmid encoding the BRET-based $G_{i1}$ ($G_{\beta1}$-2A-cpVenus-$G_{\gamma2}$-IRES-Nluc-$G_{\alpha i1}$) or $G_s$ ($G_{\beta1}$-2A-cpVenus-$G_{\gamma1}$-IRES-Nluc-$G_{\alpha s(long\ isoform)}$) sensor along with wild-type and mutant receptors or pcDNA control using Lipofectamine 2000 and seeded onto poly-D-lysine-precoated white wall, white bottomed 96-well plates. Porcupine inhibitor C59 was added 24 h after transfection to a final concentration of 10 nM. Forty-eight hours after transfection, all wells were rinsed with 150 µl HBSS and incubated with 10 µM coelenterazine-h (in HBSS) for 5 min. Subsequently, luminescence in the Nluc (450-40) and cpVenus (535–30) channel were recorded using a CLARIOstar plate reader (BMG, Ortenberg, Germany). Thereafter, cpVenus fluorescence was quantified (excitation 497-15, emission 540–20) to control for the expression levels of the G protein sensor.

**BRET-based mini Gsi recruitment assay**. ΔSMO HEK293A cells were transiently transfected in suspension using Lipofectamine®2000 (Thermo Fisher Scientific). $4 \times 10^5$ cells were transfected in 1 ml with 100 ng of C-terminally Nluc-tagged receptor plasmid DNA and 900 ng of Venus-mGsi plasmid DNA. The cells (50–100 µl) were seeded onto a poly-D-lysine-coated black 96-well cell culture plate with solid flat bottom (Greiner Bio-One). Twenty-four to forty-eight hours post-transfection, the cells were washed once with 200 µl of HBSS (HyClone). Next, 90 µl of Nluc substrate furimazine was added (1:1000 dilution; Promega) and the cells were incubated for 10 min. Subsequently, the cells were stimulated with ligands (10 µl). The BRET ratio was determined as the ratio of light emitted by Venus (energy acceptor) and light emitted by Nluc-tagged receptors (energy donors). The % ΔBRET ratio for each well is defined as: (raw BRET ratio $_{stimulated}$ − raw BRET ratio $_{basal}$)/raw BRET ratio $_{basal} \times 100$%. The vehicle corrected % ΔBRET ratio for each well is defined as: % ΔBRET ratio − average % ΔBRET ratio $_{vehicle}$. BRET acceptor (bandpass filter 535–30 nm) and BRET donor (bandpass filter 475–30 nm) emission signals were measured using a CLARIOstar microplate reader (BMG). The data presented in this study come from the ligand-induced BRET measurements obtained 5 min (FZD$_6$) or 13 min (SMO) after the ligand addition. The data were analyzed using GraphPad Prism 6.

**BRET-based cPKA recruitment assay**. ΔSMO HEK293A cells[20] were transfected in suspension with 100 ng Receptor-Nluc, increasing amounts of cPKA-YFP[44] (between 0 and 900 ng) and pcDNA to add up to a total of 1 µg plasmid DNA per ml cell suspension. Transfected cells were seeded onto PDL-precoated black-wall, black-bottomed 96-well plates (30,000 cells/well) and incubated for 48 h. Subsequently, cells were washed with 100 µl HBSS/well and incubated with 60 µl HBSS. YFP fluorescence intensity was recorded upon external excitation ($\lambda_{Ex} = 497/15$; $\lambda_{Em} = 540/20$ nm) to assess cPKA-YFP expression levels in the individual transfection samples. Next, 20 µl of 40 µM coelenterazine-H was added to all wells, incubated for 2–3 minutes, and BRET between Nluc ($\lambda_{Em} = 450/40$ nm) and YFP ($\lambda_{Em} = 535/30$ nm) was recorded in two consecutive reads (integration time 0.3 seconds). All cPKA recruitment experiments were conducted using a CLARIOstar plate reader.

**BRET-based NbSmo2 recruitment assay**. ΔSMO HEK293A cells[20] were transfected in suspension with 100 ng Receptor-Nluc, increasing amounts of NbSmo2-YFP[44] (between 0 and 900 ng) and pcDNA to add up to a total of 1 µg plasmid DNA per ml cell suspension. Transfected cells were seeded onto PDL-precoated black-wall, black-bottomed 96-well plates (30,000 cells/well) and incubated for 48 h. Subsequently, cells were washed with 100 µl HBSS/well and incubated with 60 µl HBSS. YFP fluorescence intensity was recorded upon external excitation ($\lambda_{Ex} = 497/15$; $\lambda_{Em} = 540/20$ nm) to assess NbSmo2-YFP expression levels in the individual transfection samples. Next, 20 µl of 40 µM coelenterazine-H was added to all wells, incubated for 2–3 minutes, and BRET between Nluc ($\lambda_{Em} = 450/40$ nm) and YFP ($\lambda_{Em} = 535/30$ nm) was recorded in two consecutive reads (integration

time 0.3 seconds). All NbSmo2 recruitment experiments were conducted using a CLARIOstar plate reader.

**DVL2 bystander BRET assay**. ΔFZD$_{1-10}$ HEK293 T cells[74] were transiently transfected in suspension using Lipofectamine 2000 (Thermo Fisher Scientific). $4 \times 10^5$ cells ml$^{-1}$ were transfected with 780 ng of Venus-KRas plasmid DNA, 200 ng of the SNAP-tagged receptor plasmid DNA and 20 ng of Nluc-DVL2 plasmid DNA. The cells (100 µl) were seeded onto a PDL-coated black 96-well cell culture plate with solid flat bottom (Greiner Bio-One). Twenty-four hours post-transfection, cells were washed once with HBSS (HyClone) and maintained in the same buffer. Subsequently coelenterazine-h (5 µM) was added and after 10 min incubation BRET signal was determined as the ratio of light emitted by Venus-tagged biosensors (energy acceptor) and light emitted by Nluc-tagged biosensors (energy donor). The BRET acceptor (535–30 nm) and BRET donor (475–30 nm) emission signals were measured using a CLARIOstar microplate reader (BMG). Data were analyzed using GraphPad Prism 6.

**DEP recruitment assay**. ΔFZD$_{1-10}$ HEK293 T cells were transiently transfected in suspension using Lipofectamine 2000 (Thermo Fisher Scientific). $4 \times 10^5$ cells ml$^{-1}$ were transfected with 500 ng of Venus-KRas plasmid DNA, 0–480 ng of the SNAP-tagged receptor plasmid DNA, 20 ng of Nluc-DEP plasmid DNA and pcDNA to a final amount of 1 µg of DNA. The cells (100 µl) were seeded onto a PDL-coated black 96-well cell culture plate with solid flat bottom (Greiner Bio-One). Twenty-four hours post-transfection, cells were washed once with HBSS (HyClone) and maintained in the same buffer. Subsequently coelenterazine-h (5 µM) was added and after 10 min incubation BRET signal was determined as the ratio of light emitted by Venus-tagged biosensors (energy acceptor) and light emitted by Nluc-tagged biosensors (energy donor). The BRET acceptor (535–30 nm) and BRET donor (475–30 nm) emission signals were measured using a CLARIOstar microplate reader (BMG). Cells were washed once with HBSS and incubated with 50 µl of 1 µM SNAP-surface Alexa Fluor 647 (New England Biolabs, #S9136S) in complete DMEM medium for 30 min at 37 °C and 5% $CO_2$. Subsequently, cells were washed three times with HBSS and the fluorescence (excitation 625–30 nm, emission 680–30 nm) was read with a CLARIOstar microplate reader (BMG). Data were analyzed using GraphPad Prism 6.

**TOPFlash assay**. $6 \times 10^5$ cells ml$^{-1}$ ΔFZD$_{1-10}$ HEK293 T cells were seeded onto PDL-coated white 96-well cell culture plate with solid flat bottom (Greiner Bio-One). Next day, cells were transfected with 20 ng of SNAP-tagged receptor, 20 ng M50 Super 8× TOPFlash (Addgene, 12456), 2 ng pRL-TK Luc (Promega, E2241) and 58 ng pcDNA plasmid DNA to a final amount of 100 ng of plasmid DNA per well. Four hours after transfection, medium was changed to starvation medium (DMEM without FBS) containing either vehicle or 1000 ng/ml recombinant WNT-3A for FZD transfected cells or 100 nM SAG1.3 for SMO-transfected cells. Twenty-four hours after stimulation, cells were lysed gently shaking with 20 µl 1× Passive Lysis Buffer (Promega, E1910) for 15 min. Subsequently, 20 µl of LAR II (Promega, E1910) were added to all wells after which luminescence (580-80 nm) was read and then 20 µl of Stop & Glo (Promega, E1910) were added to all wells after which luminescence (480-80 nm) was read again with a CLARIOstar microplate reader (BMG). Data were analyzed using GraphPad Prism 6.

**SNAP-surface Alexa Fluor 647 staining**. For quantification of cell-surface expression of N-terminally SNAP-tagged receptors, ΔFZD$_{1-10}$ HEK293T cells at the density of $4 \times 10^5$ cells ml$^{-1}$ were transfected in suspension using Lipofectamine 2000 with 200 ng of the indicated SNAP-tagged receptor plasmid DNA and 800 ng of the pcDNA plasmid DNA. The cells (100 µl) were seeded onto a PDL-coated black 96-well cell culture plate with solid flat bottom (Greiner Bio-One). 24 h later the cells were washed once with HBSS (HyClone) and incubated with 50 µl of 1 µM SNAP-surface Alexa Fluor 647 (New England Biolabs, #S9136S) in a complete DMEM medium for 30 min at 37 °C. Subsequently, the cells were washed three times in HBSS and the fluorescence (excitation 625–30 nm, emission 680–30 nm) was read with a CLARIOstar microplate reader (BMG). Data were analyzed using GraphPad Prism 6.

**ELISA surface expression**. For quantification of cell-surface expression of C-terminally Nluc-tagged receptors, ΔFZD$_{1-10}$ HEK293T or ΔSMO HEK293A cells at the density of $4 \times 10^5$ cells ml$^{-1}$ were transfected in suspension using Lipofectamine 2000 with either 500 ng of the indicated FLAG-tagged FZD$_6$-Nluc receptor plasmid DNA and 500 ng of the pcDNA plasmid DNA or 1 µg of FLAG-SNAP-tagged SMO-Nluc receptor plasmid DNA (Supplementary Fig. 6f). The cells (100 µl) were seeded onto a PDL-coated transparent 96-well cell culture plate with solid flat bottom. Twenty-four hours later the medium was dispensed from the wells and washed once with 200 µl of ice-cold wash buffer (0.5% BSA in PBS), after which cells were incubated on ice with 25 µl of primary antibody solution (1% BSA in PBS with anti-FLAG 1:500 (Sigma–Aldrich F1804)) for 1 h. Subsequently, cells were washed as above four times and then incubated on ice with 50 µl of secondary antibody solution (1% BSA in PBS with HRP-conjugated anti-mouse 1:3000 (Invitrogen 31430)) for 1 h, after which cells were washed as above four times. Lastly, 50 µl of TMB (3,3′,5,5′-Tetramethylbenzidine, Sigma T0440) was added to

each well and incubated for 20 min after which 50 µl of 2 M HCl was added and incubated for 20 min. Absorbance (450 nm) was read with a POLARstar Omega microplate reader (BMG). Data were analyzed using GraphPad Prism 6.

**Data handling and analysis**. Plate readers were operated using Reader Control 5.21 R4 and Mars 3.20 R2 software for the BMG CLARIOstar plate reader and Omega 1.10 and MARS 1.11 software for the BMG POLARstar Omega plate reader. Excel 2007 or 2013 was used for saving raw data and for data transfer to GraphPad Prism 6, Raw BRET was defined as acceptor emission intensity over donor emission intensity. For the NanoBiT/BRET-binding data presented, the saturation binding curves were fitted using a three- or four-parameter nonlinear regression model. The binding curves represent mean ± standard error of the mean (SEM) from three independent experiments, each performed in two technical replicates. Affinity values are presented as a best-fit $K_d$ ± standard deviation (SD). NanoBiT/BRET-binding models were selected based on an extra sum-of-square F-test ($P < 0.05$). Raw BRET ratios of the G protein sensor experiments were normalized to the pcDNA control to account for interday BRET variability. BRET-based mGsi recruitment data were fitted to a bell-shaped model.

To analyze the cPKA- and NbSmo2-YFP recruitment data, YFP fluorescence intensities (upon external excitation) and raw BRET ratios of each transfection ratio (four replica) were averaged, pooled from all individual experiments and plotted as mean ± SD. Each dataset was then fitted to a one-phase association or linear model and the preferred fit was selected based on an Extra sum-of-squares F-test ($P < 0.05$). Extra sum-of-squares F-test was further applied to compare the BRET plateaus of SMO-Nluc wild-type and SMO-Nluc $F^{6.43}P$ in the cPKA-YFP recruitment assay ($P < 0.05$). For the Nluc-DVL2 recruitment data, the net BRET values (bystander BRET) for each receptor subtype were normalized to the corresponding values of the normalized surface expression (Supplementary Fig. 6e). Where applicable, data were analyzed for differences with one-way ANOVA with Fisher's least significant difference post-hoc analysis or paired $t$-test. Significance levels are given as $*P < 0.05$; $**P < 0.01$; $***P < 0.001$; $****P < 0.0001$. Please refer to the figure legends for more details on the displayed data.

**Reporting summary**. Further information on research design is available in the Nature Research Reporting Summary linked to this article.

## Data availability

Data supporting the findings of this manuscript are available from the corresponding author upon reasonable request. Full molecular dynamics simulations are available at the open access database GPCRmd (www.gpcrmd.org; simulation system IDs 239–244). Snapshots of the receptor models presented in this study (extracted from the MD trajectories) are provided as Supplementary Data. Source data is provided with this article. Source data are provided with this paper.

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

## Acknowledgements

We thank Anna Krook for access to the CLARIOstar plate reader and Benoit Vanhollebeke for the ΔFZD$_{1-10}$ HEK293T cells. The work was supported by grants from Karolinska Institutet, the Swedish Research Council (2017-04676; 2019-01190), the Swedish Cancer Society (CAN2017/561, 20 1102 PjF, 20 0264 P), the Novo Nordisk Foundation (NNF17OC0026940, NNF20OC0063168), Wenner-Gren Foundations (UPD2018-0064; UPD2019-0193), The Lars Hierta Memorial Foundation (FO2019-0086, FO2020-0304), The Alex and Eva Wallström Foundation for scientific research and education (2020-00228), The Swedish Society of Medical Research (SSMF; P19-0055), and the German Research Foundation (DFG; 427840891). Computational resources were provided by the Swedish National Infrastructure for Computing—National Supercomputer Centre (NSC) in Linköping and KTH Royal Institute of Technology (PDC) in Stockholm (SNIC 2020/5-500).

## Author contributions

A.T. and G.S. conceived and designed the study. A.T. performed the in silico work. P.K., H.S., and C.-F.B. performed the wet lab experiments. A.T., P.K., H.S., C.-F.B., and G.S. designed and prepared the figures. A.T. and G.S. wrote the manuscript. P.K., H.S., and C.-F.B. commented and contributed to the manuscript writing. G.S. supervised and coordinated the project.

## Funding

## Competing interests

The authors declare no competing interests.
