## [Peer Review File · Nature Communications]

Reviewers' Comments:

Reviewer #1:

Remarks to the Author:

The manuscript by Turku et al. is an elegant multidisciplinary study of activation mechanisms of Class F G protein coupled receptors (GPCRs) exemplified by the Smoothed receptor (SMO) and ten Frizzled receptors (FZD). The authors uncover a role for a conserved proline residue in TM6 of these receptors, and the helical kink induced by that proline, as well as unique differences between SMO (where this proline is not present and the corresponding residue is a Phe) and FZD's. Homology modeling, molecular dynamics, and BRET-based assays probing antagonist binding and the recruitment of multiple effectors and probes to WT and mutated receptors in live cells are used to substantiate the findings.

Major comment

The study is rigorous and features a wide range of clever molecular sensors to thoroughly characterize constitutive signaling of the receptors. One important aspect that is missing, and that would greatly strengthen the paper in my view, would be characterization of mutation effects on ligand-induced signaling in addition to that. The authors have at hand a ligand, SAG1.3, that is a known agonist of SMO and that they have previously demonstrated to be an agonist of FDZ6 as well. Nevertheless, in the present manuscript, agonist-induced signaling is not discussed at all. How do mutations affect agonist binding and the associated conformational changes? What role does the proline residue play, and how are SMO and FZD's different in mechanisms of agonist-induced signaling? I realize that HEK293 cells may not be the best system for studying agonist-induced signaling, due to lacking regulatory mechanisms and, consequently, a high level of constitutive activity of the receptors in these cells; however, this omission makes the message of the paper somewhat incomplete.

Minor comments

* Intro: "active, G protein-coupling or nanobody-stabilized, high resolution structures were resolved for tens of different receptors of class A and class B, and one representative of class F." – I think it is also important to mention the availability of inactive state structures. Also, I don't think there are dozens of Class B structures, so the statement seems a little sloppy. There are also structures of Class C receptors.

* In the simulations, was the ring puckering on the proline in question up or down? (not to be confused with cis- and trans-proline; I realize that it was trans). The puckering of the proline ring may have profound effects on the stability and conformation of the helix containing it, and if not sampled adequately, may lead to artefacts.

* "Furthermore, the unaffected BODIPY-cyclopamine binding in FZD6 P6.43F compared to the wild type receptor indicated that straightening out TM6 in FZD6 neither affected the upper nor uncovered a lower binding site." – why is it not a possibility the lower binding site is either affected or uncovered, but just does not bind cyclopamine?

* Discussion is excessively detailed and somewhat repetitive in comparison with the Results section.

Reviewer #2:

Remarks to the Author:

Turku et al. demonstrate that FZDs form a kinked TM6 upon activation through computational modeling, MD simulation and BRET assays, unlike SMO, which showed the outward movement of TM6 without helical bending upon activation. The authors suggested that the difference in activation-dependent conformational changes between SMO and FZDs is due to the residue at position 6.43 in TM6 (Pro vs. Phe). These results provide a better understanding of the activation mechanism of class F GPCRs.

There are some issues which will need to be clarified before publishing will be considered. Several of

them are listed below and the most critical one is shown here.

While SMO mediates hedgehog signaling, FZDs mediate WNT signaling in response to various subtypes of WNTs. However, the authors have not observed FZD activation by WNT and have not demonstrated the importance of TM6 bending at P6.43 during WNT signaling. MD simulation was performed with the systems stimulated by SAG1.3 and constitutive Gi activity and beta-arrestin2 recruitment at the basal level were monitored by BRET assays. Also, constitutive DVL2 recruitment assays using WT and mutants FZDs were carried out by bystander BRET assay. I am not convinced that these results support the activation mechanism of FZDs.

Moreover, in these assays, FZD subtypes showed different results. The FZD6 P6.43F mutation slightly reduced Gi activity and significantly reduced DVL2 recruitment. On the other hand, the FZD7 P6.43F mutant did not show significant differences in Gs activity and DVL2 recruitment. In the discussion section, the authors tried to explain these differences with different G protein subtype interaction and different WNT signaling pathway. It is likely that the activated FZDs form a kinked TM6 at the conserved P6.43 in G protein signaling (similar to the class A GPCRs-G protein complexes), but it is difficult to generalize "FZD activation mechanism" from these results, because there is insufficient data for the interaction between FZDs and DVL2 upon WNT activation. Nevertheless, the authors concluded that "FZDs rely on residue P6.43 enabling a kinked TM6 upon activation---" in the abstract. It is certainly interesting that the P6.43 residue of FZDs would play an important role in TM6 bending upon activation (in some cases), however, I am not convinced that the author's investigation clearly explains the FZD activation mechanism enough to justify publication in Nature Communications.

Additional issues:

The authors emphasize that FZDs and SMO have structural similarities in both transmembrane ligand and intracellular effector binding sites (page 3), but have different conformational changes upon activation depending on the presence of Pro at position 6.43 in TM6. However, this statement seems to over-emphasize the structural similarity between FZDs and SMO. Although SMO and FZDs belong to the same class F GPCR family, they show structural and functional differences. First of all, since DVL cannot bind to SMO, it is difficult to say that FZDs and SMO show structural similarity at the intracellular effector binding sites. Second, SMO has an extended TM6, which is a peculiar structural aspect not found in other GPCRs. In contrast, FZD4 and FZD5 do not have an extended TM6, and instead form a short alpha-helix and a flexible loop at the corresponding region of SMO (ECL3). The authors used the previously published homology model of FZD6, which was built using FZD4 and SMO structures as templates (published in Nat. Commun. 2020 by the same group). Now, the cryo-EM structure of FZD5, which has longer ECL3 than FZD4 is available. A homology model of FZD6 should be updated using FZD4 and FZD5 structures as templates and loop modeling. Third, the inactive structures of FZD4/5 and SMO showed structural difference in the C-terminal part of TM7 as well. Unfortunately, the authors did not describe the structural differences between FZD4/5 and SMO (in an inactive state).

Figure 2B. The authors showed that F6.43P mutation (in SMO) significantly reduced the BODIPY-cyclopamine binding affinity, and explained this result with a decrease in the volume of the lower pocket (high affinity binding pocket) in this mutant (page 6). To support this authors' explanation, it would be nice if they could show that this mutant does not show a difference in affinity for the upper binding site (using Δ CRD-SMO or SAG 1.3).

Page 4. The authors wrote that "FZD6 was selected for the MD simulations due to its similarity to SMO". It is not clear in what aspects the authors refer to similarity. Please clarify this.

Supplementary Figure 2. The FZD7 mutant appears to have lower surface expression than the wild type, but it says that the difference is not significant. What is the p-value? Which t-test was used? There is not enough information about statistical analysis of data throughout this manuscript.

Minor issues

Figure 1A. In the figure of class A GPCRs, aligning the structures of the C-terminal part of TM6 will show the outward movement of the cytoplasmic part of TM6 more clearly. Also, light grey labeling is hard to see.

Page 7. The authors wrote that "The only FZD crystal structures currently available are FZD4 and FZD5". However, the FZD5 structure is not a crystal structure but a Cryo-EM structure.

Page 17. Recent findings that SMO-tagerting small => SMO-targeting

Reviewer #3:

Remarks to the Author:

In this study, Turku et al. conduct molecular dynamics simulations and numerous BRET assays to investigate the mechanism of activation of Class F GPCRs. Class F GPCRs, which consist of the Frizzled (FZD) and Smoothed (SMO) receptors, have important physiological roles including in cell polarity, cell proliferation and embryonic development. The authors find an important role for the residue at position 6.43 (P in FZD receptors, F in SMO receptors) in FZD/SMO activation. Mutagenesis of this position has various implications including on TM6 conformation, ligand binding, G protein-coupling and beta-arrestin2 recruitment.

This study presents novel findings that shed light on the activation mechanisms of Class F GPCRs, which are currently not well understood. These results will have important consequences for drug design and our understanding of GPCR structure and activation more broadly. The manuscript encompasses a comprehensive set of experiments and the results are generally explained and presented well; the figures are clear and very visually appealing. I support the publication of this manuscript with minor revisions, as outlined below.

Major comments

1. A more comprehensive sequence alignment could be generated, for example using ortholog sequences obtained from Ensembl Compara (there are at least >100, and for most receptors >200, one-to-one ortholog sequences for each Class F receptor in Ensembl). An alignment built from sequences from a larger number of species would give more robust indications of how conserved certain amino acids/positions are.
2. Many important GPCR-lipid interactions have been identified, including ones that influence activation mechanisms and TM6 conformations. Could the authors comment on their choice to use a pure POPC bilayer (and not include other lipids found in eukaryotic/mammalian membranes) in the MD simulations?
3. The text states that the MD simulations of active(-like) FZD6 and SMO ran for 1250 ns, but Fig 1e of the TM6 angles only shows 500 ns.
4. Are the differences in binding cavity volume between FZD/SMO WT and mutant structures statistically significant (Fig 2d)?
5. Could the authors comment on why they did not normalize the results in the Gi-BRET sensor assay for receptor surface expression (Fig 3b)?

6. Could the authors comment on why they used a bystander BRET assay, rather than a BRET assay like the one used to measure cPKA/NbSmo2 recruitment, to measure DVL2 recruitment? With respect to this, the inclusion of a negative control could be important to confirm that the BRET signal from the bystander BRET assay represents specific binding of DVL2 to FZD receptors.

7. The discussion of DVL2 recruitment to FZD7 is slightly confusing: the cell surface expression-normalized results are not statistically significant (Fig 5f), so claiming that there is a "similar trend" of reduced DVL2 recruitment using the non-normalized data from the supplementary figure (Fig S7) seems misleading.

8. Could the authors comment on how including the mini-Gi protein (as done for simulations presented in the Discussion section) would influence the binding cavity volume in the active-like FZD simulations (as presented in Fig 2d)?

9. Figures are labelled with * and ** – could the authors clarify what (presumably p-values) these refer to?

Minor comments

1. More detail on the "similarity" between FZD6 and SMO, which factored into the choice to use FZD6 for the MD simulations, would be helpful (e.g. sequence similarity and if so, what percent?).

2. The FZD6-SAG1.3-miniGi simulations are first mentioned in the Discussion. It would be helpful to move these results to the Results section.

Figure comments

1. Fig 1b: the top panel (appears to be a measure of sequence conservation) is not explained in the figure legend.

2. Fig 1e: it appears as though replica 4 is starting halfway through (at 250 ns). I am not sure if this is merely a result of the data representation, but it is slightly confusing.

This study demonstrates significant and novel findings, which will advance our understanding of Class F GPCR activation and have important implications for drug design. I enjoyed reading this work and support its publication with the relevant revisions.

Alissa M Hummer

Comments from the authors are presented in green; addition of new data is highlighted in **bold**.

REVIEWER COMMENTS

Before going into detailed responses to the reviewers, who all provided highly appreciated and constructive criticism, we would like to summarize the additional data that were produced for this revision:

1. **New Supplementary Figure 1:** extract of a recently published (Wright and Koziellewicz et al 2019, Nature Communications) mega alignment for class F receptors providing information of the phylogenetic conservation of the P/F^{6.43} in FZD/SMO over ca 750 different species.
2. **New Supplementary Figure 3:** analysis of the MD with regard to proline puckering in FZD₆ and SMO F^{6.43}P
3. **New Supplementary Figure 5:** BODIPY-cyclopamine binding in Δ CRD SMO wild type and F^{6.43}P showing that the upper binding site of BODIPY-cyclopamine that is uncovered by CRD removal (as reported in Koziellewicz et al 2020, Molecular Pharmacology) is not affected by mutating F^{6.43}P.
4. **New Supplementary Figure 11:** Comparison of SAG1.3-induced mini Gi protein recruitment to SMO and FZD₆ wild type and corresponding residue 6.43 mutant.
5. **New Fig. 6c:** DEP recruitment to FZD₆ wild type and P^{6.43}F using a BRET-based assay that controls for cell surface expression confirming the strong effect of the mutation in FZD₆ reported for constitutive full-length DVL recruitment (Fig. 5f of the original manuscript).
6. **New Fig. 6d:** Comparative TOPflash assay comparing the ligand-induced signaling ability of wild type and 6.43 mutant in FZD_{4, 5, 6, 7}, and SMO.

Reviewer #1 (Remarks to the Author):

The manuscript by Turku et al. is an elegant multidisciplinary study of activation mechanisms of Class F G protein coupled receptors (GPCRs) exemplified by the Smoothed receptor (SMO) and ten Frizzled receptors (FZD). The authors uncover a role for a conserved proline residue in TM6 of these receptors, and the helical kink induced by that proline, as well as unique differences between SMO (where this proline is not present and the corresponding residue is a Phe) and FZD's. Homology modeling, molecular dynamics, and BRET-based assays probing antagonist binding and the recruitment of multiple effectors and probes to WT and mutated receptors in live cells are used to substantiate the findings.

We appreciate the reviewer's positive comments and are happy that this reviewer appreciated our findings.

Major comment

The study is rigorous and features a wide range of clever molecular sensors to thoroughly characterize constitutive signaling of the receptors. One important aspect that is missing, and that would greatly

strengthen the paper in my view, would be characterization of mutation effects on ligand-induced signaling in addition to that. The authors have at hand a ligand, SAG1.3, that is a known agonist of SMO and that they have previously demonstrated to be an agonist of FZD6 as well. Nevertheless, in the present manuscript, agonist-induced signaling is not discussed at all. How do mutations affect agonist binding and the associated conformational changes? What role does the proline residue play, and how are SMO and FZD's different in mechanisms of agonist-induced signaling? I realize that HEK293 cells may not be the best system for studying agonist-induced signaling, due to lacking regulatory mechanisms and, consequently, a high level of constitutive activity of the receptors in these cells; however, this omission makes the message of the paper somewhat incomplete.

The reviewer raises an important point. However, when it comes to ligand-induced changes, we encounter more diverse factors than when assessing constitutive activities since ligand potencies/efficacies and receptor ligand-selectivities and receptor-signaling pathway selectivities differ, which makes it difficult to compare. However, in order to address this reasonable and clever point, we have added:

1. Comparison of the SAG1.3-induced miniGsi recruitment for SMO wt and F^{6.43}P and FZD₆ wt and P^{6.43}F mutant. (New Supplementary Figure 11)
2. Agonist-induced activation of TOPflash in Δ FZD₁₋₁₀ KO cells using FZD_{4,5,6,7} (WNT-3A), SMO (SAG1.3) - FZD₆ and SMO do not induce TOPflash. **New Fig. 6d**

The comparison of agonist (SAG1.3) binding at SMO and FZD₆ cannot be done in a meaningful way because the tracer (BODIPY-cyclopamine) is affected differently by the mutation in FZDs vs SMO. Agonist binding (SAG1.3) needs to be done in a competition binding setup using a fixed BODIPY-cyclopamine concentration and competition with SAG1.3. However, the large effect of the F to P mutation in residue 6.43 make is almost impossible to draw firm conclusions comparing FZD and SMO.

This figure is included for the reviewers information only here in the rebuttal letter:

We hope that these approaches complement the manuscript sufficiently.

Minor comments

* Intro: "active, G protein-coupling or nanobody-stabilized, high resolution structures were resolved for tens of different receptors of class A and class B, and one representative of class F." – I think it is also

important to mention the availability of inactive state structures. Also, I don't think there are dozens of Class B structures, so the statement seems a little sloppy. There are also structures of Class C receptors.

Thanks for this comment and we agree with the reviewer. The statement has been refined and also includes a reference to inactive state structures now.

* In the simulations, was the ring puckering on the proline in question up or down? (not to be confused with cis- and trans-proline; I realize that it was trans). The puckering of the proline ring may have profound effects on the stability and conformation of the helix containing it, and if not sampled adequately, may lead to artefacts.

We thank the reviewer for this highly relevant comment and ask him/her to refer to the **new Supplementary Figures 3 and 4** for further insights into the ring conformations throughout the simulation trajectories. In Fig. S3 and S4, we present the χ_2 torsion angles of the proline residues in FZD₆ wild type (\pm miniG_i) and SMO F^{6.43}P simulations, as the χ_2 angle has been reported to be a reliable single measure of the ring puckering states of the proline (Wu, AIP Advances 3, 032141, 2013 <https://doi.org/10.1063/1.4799082>). In the FZD₆ simulations, the proline 6.43 is in an up-puckering state in approximately 85% of the simulation frames, which is in line with the puckering state observed in the prolines in the middle of alpha helices in PDB (Milner-White et al., JMB 228, 725-734, 1992; [https://doi.org/10.1016/0022-2836\(92\)90859-1](https://doi.org/10.1016/0022-2836(92)90859-1)). As the up-puckering state is the one allowing more relaxed (i.e. less kinked) alpha helix conformation, we are confident that our FZD₆ simulations are not over-estimating the kink.

In SMO F^{6.43}P simulations the proline puckering conformations are approximately equally present (51% in up and 49% in down conformation). Interestingly, this is in line with observations of those *trans*-prolines in PDB that are not located in alpha helices (Milner-White et al., JMB 228, 725-734, 1992; [https://doi.org/10.1016/0022-2836\(92\)90859-1](https://doi.org/10.1016/0022-2836(92)90859-1)). Why the P^{6.43} the SMO F^{6.43}P system resembles more the non-helical proline data might be due to the presence of the glycine residue at position 6.42 of SMO (i.e. next to the studied proline towards intracellular side of the helix), which, due to lack of a side chain, provides more space for proline residue to access the down-puckering state than the corresponding valine residue does in FZD₆. Despite this additional flexibility, the SMO simulations are sampling both puckering states throughout the simulation trajectories, not only one of them.

* “Furthermore, the unaffected BODIPY-cyclopamine binding in FZD6 P6.43F compared to the wild type receptor indicated that straightening out TM6 in FZD6 neither affected the upper nor uncovered a lower binding site.” – why is it not a possibility the lower binding site is either affected or uncovered, but just does not bind cyclopamine?

With this sentence we indeed meant that straightening out the TM6 of FZD₆ did not make the BODIPY-cyclopamine to bind to the lower pocket of the receptor – as it does not do in the wild type FZD₆ either. We have reworded the sentence as follows: “Furthermore, the unaffected BODIPY-cyclopamine binding in FZD₆ P^{6.43}F compared to the wild type receptor indicated that straightening out TM6 in FZD₆ neither affected the upper binding site nor enabled the lower binding site to bind BODIPY-cyclopamine.”

* Discussion is excessively detailed and somewhat repetitive in comparison with the Results section.

We have gone through the discussion and removed or shortened parts that appeared repetitive.

Reviewer #2:

Turku et al. demonstrate that FZDs form a kinked TM6 upon activation through computational modeling, MD simulation and BRET assays, unlike SMO, which showed the outward movement of TM6 without helical bending upon activation. The authors suggested that the difference in activation-dependent conformational changes between SMO and FZDs is due to the residue at position 6.43 in TM6 (Pro vs. Phe). These results provide a better understanding of the activation mechanism of class F GPCRs.

There are some issues which will need to be clarified before publishing will be considered. Several of them are listed below and the most critical one is shown here.

While SMO mediates hedgehog signaling, FZDs mediate WNT signaling in response to various subtypes of WNTs. However, the authors have not observed FZD activation by WNT and have not demonstrated the importance of TM6 bending at P6.43 during WNT signaling. MD simulation was performed with the systems stimulated by SAG1.3 and constitutive Gi activity and beta-arrestin2 recruitment at the basal level were monitored by BRET assays. Also, constitutive DVL2 recruitment assays using WT and mutants FZDs were carried out by bystander BRET assay. I am not convinced that these results support the activation mechanism of FZDs.

Moreover, in these assays, FZD subtypes showed different results. The FZD6 P6.43F mutation slightly reduced Gi activity and significantly reduced DVL2 recruitment. On the other hand, the FZD7 P6.43F mutant did not show significant differences in Gs activity and DVL2 recruitment. In the discussion section, the authors tried to explain these differences with different G protein subtype interaction and different WNT signaling pathway. It is likely that the activated FZDs form a kinked TM6 at the conserved P6.43 in G protein signaling (similar to the class A GPCRs-G protein complexes), but it is difficult to generalize “FZD activation mechanism” from these results, because there is insufficient data for the interaction between FZDs and DVL2 upon WNT activation. Nevertheless, the authors concluded that “FZDs rely on residue P6.43 enabling a kinked TM6 upon activation---” in the abstract.

It is certainly interesting that the P6.43 residue of FZDs would play an important role in TM6 bending upon activation (in some cases), however, I am not convinced that the author’s investigation clearly explains the FZD activation mechanism enough to justify publication in Nature Communications.

The reviewer’s comments are indeed important, and we would like to underline that the term “FZD activation” describes a complex phenomenon and presents most likely also a matter of intense debate with the differences of receptor activation in a signalosome feeding into WNT/ β -catenin signaling vs. FZD activation as GPCR at heart.

In the revised version of the manuscript, we have attempted to clarify this better and to soften out the wording to avoid overstatements. Furthermore, we would like to refer to the title in a recent Nature Communication paper by our group (Wright and Kozielowicz et al 2019): A conserved molecular switch in

Class F receptors regulates receptor activation and pathway selection. In that paper we report on the molecular switch mechanism and its role in the agonist-induced miniG protein recruitment. There receptor activation referred to activity as a GPCR.

In that regard the current study adds additional structure-function data that support our claim that the bent TM6 contributes to FZD activation or activation-associated conformational changes.

We would also like to underline that substantial data emerge currently – in part published, though mostly unpublished (including this manuscript) – that paralogues of FZDs present with a relatively high constitutive (i. e, ligand-independent) activity towards heterotrimeric G proteins. This is particularly evident in Fig. 3 presenting the G protein activation through wild type and P^{6.43}F FZD₆ and FZD₇. In addition, we have more detailed evidence for the full class F in unpublished data, which might have influenced our wording unconsciously, efforts that basically presents the continuation of our work in Wright and Kozielowicz et al 2019, Nature Communications, where we had seen tendency in wild type FZD₆ to present with constitutive activity, but which could not be resolved at the time given the assays at hand.

However, we would also like to underline that our previous findings using intramolecular FRET probes for FZD₅ (Wright et al Science Signaling 2018, PMID 30514810) and miniG proteins as conformational sensors for the active state (TM6 swing out; Wright and Kozielowicz et al 2019, Nature Communications, PMID 30737406) of FZDs as well as the very recently published paper (Schihada et al 2020

<https://doi.org/10.1016/j.bios.2020.112948> reporting on conformational sensors for FZD_{4, 5, 6, 7}) clearly show that WNT-induced activation indeed involves conformational changes that are in agreement with our statement in the current manuscript. These findings emphasize on a broader scale for class F that conformational changes involving rearrangements reminiscent of Class A/B GPCRs are elicited by WNTs.

The reviewer is completely correct that little is known about dynamic DVL recruitment to FZDs in response to WNTs. This in fact is a central question in the Schulte laboratory, on which we are working intensively. Nevertheless, it is obvious that FZDs that do recruit DVL (in the absence of WNTs, i.e. constitutive recruitment) can mediate WNT/ β -catenin signaling and those that e.g. are impaired by mutations cannot. However, FZD-DVL interaction *per se* is not sufficient to mediate WNT/ β -catenin signaling as exemplified by FZD₆, which recruits DVL but does not activate WNT/ β -catenin signaling. To clarify the ability of the FZDs under investigation to mediate WNT-3A-induced TOPflash, we have used wt receptors and their P^{6.43}F mutants in Δ FZD₁₋₁₀ KO HEK293 cells. These data are now presented in **Fig. 6d**. Along a similar line, we added a more detailed analysis of FZD₆-DVL DEP interaction in **Fig. 6c** arguing that FZD conformation by mutation of the residue 6.43 affects the overall receptor conformation relevant for DEP recruitment.

Since FZD₆ does not mediate WNT/ β -catenin signaling, we complement this functional data set with a SAG1.3-induced and FZD₆-dependent recruitment of miniGsi proteins (similar to what was presented in Wright and Kozielowicz et al 2019, Nature Communications). See **New Supplementary Figure 11**.

Additional issues:

The authors emphasize that FZDs and SMO have structural similarities in both transmembrane ligand and intracellular effector binding sites (page 3), but have different conformational changes upon activation depending on the presence of Pro at position 6.43 in TM6. However, this statement seems to over-

emphasize the structural similarity between FZDs and SMO. Although SMO and FZDs belong to the same class F GPCR family, they show structural and functional differences. First of all, since DVL cannot bind to SMO, it is difficult to say that FZDs and SMO show structural similarity at the intracellular effector binding sites. Second, SMO has an extended TM6, which is a peculiar structural aspect not found in other GPCRs. In contrast, FZD4 and FZD5 do not have an extended TM6, and instead form a short alpha-helix and a flexible loop at the corresponding region of SMO (ECL3). The authors used the previously published homology model of FZD6, which was built using FZD4 and SMO structures as templates (published in Nat. Commun. 2020 by the same group). Now, the cryo-EM structure of FZD5, which has longer ECL3 than FZD4 is available. A homology model of FZD6 should be updated using FZD4 and FZD5 structures as templates and loop modeling. Third, the inactive structures of FZD4/5 and SMO showed structural difference in the C-terminal part of TM7 as well. Unfortunately, the authors did not describe the structural differences between FZD4/5 and SMO (in an inactive state).

Thank you very much for these comments.

In a previous paper (Wright and Koziellewicz et al 2019, Nature Communications), we have published a class F wide sequence alignment, which underlines the closer relationship of SMO with FZD₆ (compared to other FZDs), where SMO and FZD_{3/6} actually emerge from the same node in the phylogenetic tree. Other FZDs, however, emerge from different nodes of the tree (see the phylogenetic tree for class F to the right).

Furthermore, in Koziellewicz et al. 2020 (Nature Communications), we further highlighted the similarity of SMO to FZD₆ based on sequence alignments in the core of the receptor, where SAG1.3 binds.

In addition, the same studies clarify that (based on the FZD apo structure) FZD₄ cannot bind SAG1.3 because the receptor presents with a shorter TM6 that does not extend straight above the membrane but rather traverses through the area, where SAG1.3 could bind. However, the conclusion mentioned by this reviewer that FZD₅ is lacking an extended TM6 should not be drawn from the recently published FZD₅ apo structure. The extracellular domains of that structure are poorly defined and the completely conserved linker/ECL1 cysteine tetrad existing in class F (see also Valnohova et al 2018, J Biol Chem; PMID: 30237173), which are known to establish two disulphide bonds providing rigidity to the linker domain through stabilization of an antiparallel beta sheet, are not involved in disulphide bonds in the structure. Figure from Valnohova et al 2018, J Biol Chem:

The linker domain of Frizzled 6

Figure 5. Identification of a well-conserved triad of cysteines in the linker domain. *A*, close-up of the SMO structure shown in Fig. 1*A*. CRD, yellow; linker, green; 7TM core, gray. In addition, the linker domain cysteines corresponding to human FZD₆ Cys-161, Cys-181, and Cys-185 and Cys-260 in ECL1 are shown as red sticks. *B*, the table summarizes information about the presence of the antiparallel β -sheet in the linker domain from all published SMO crystal structures and one FZD₄ structure (PDB code 6BD4). FL, CRD present; Δ CRD, CRD absent. *C* and *D*, alignment of the extracellular linker domains and ECL1 of all human class F receptor homologs shows a high degree of conservation among the cysteines in the linker domain (*C*) and the cysteine in ECL1 (*D*). The bar graphs show the degree of conservation between the compared sequences. Numbers identify Cys-161, Cys-181, and Cys-185 in human FZD₆. Increasing intensity of blue indicates a higher degree of conservation. Alignment was done using MAFFT with default settings. Structures were rendered using PyMOL (PyMOL Molecular Graphics System, version 2.0, Schrödinger, LLC).

Furthermore, we have previously identified an interaction network in both SMO and FZD₆, which stabilizes the kinked extended TM6 in the SMO structures and the FZD₆ model. Thus, it is very likely – even in the absence of *de facto* structural information for FZD₆ – to assume that this paralogue presents with a similar extended TM6 as SMO. Figure from Kozielwicz et al 2020, Nature Communications Supplementary Figure 19:

In our current study, we utilize two FZD₆ models – one reflecting the active receptor conformation and another the inactive receptor conformation. The active-like model is based on the active conformation of SMO, as other active class F structures are currently not available. The sequence identity between Δ CRD-SMO and Δ CRD-FZD₆ is approximately 29% and sequence similarity approximately 48%, which are enough to allow reliable homology modeling of a GPCR structure. This active-like FZD₆ model is used in all molecular dynamics studies we present here, as we are indeed studying the active receptor conformations, and the non-accelerated all-atom molecular dynamics simulations we utilize here are not, as a method, providing the possibility to capture the transitions from the fully inactive to the fully active receptor conformations within the simulation time-scales that would be currently computationally feasible. This means that we are more likely to sample relevant conformational space of the receptor when we build the model based on the active conformation of the sufficiently similar SMO structure than the more closely related but inactive/apo FZD₄ or FZD₅ structures. The active-like FZD₆ model that we utilize here is originally published in Kozielwicz et al. 2020, Nature Communications (referred also there as an active-like FZD₆ model).

The inactive FZD₆ model instead (originally published in Schihada et al. 2020 <https://doi.org/10.1016/j.bios.2020.112948>), is based on the apo FZD₄ and ligand-bound SMO structures, of which SMO alone is used for the parts of FZD₆ for which we have shown before that FZD₄ does not offer a good template (i.e. the receptor areas involved in the SAG1.3 binding site, including the extracellular part of TM6). However, we are not using the whole extended TM6 of the template SMO structure as basis for the TM6 of FZD₆, but cutting it approximately two helical turns shorter, because the FZD₆ sequence has a proline residues pattern there that is typically found at the ends of the alpha helices of the proteins. Originally FZD₄ was used as a template instead of FZD₅ purely because the publication of the FZD₅ structure became available only when the current manuscript was already in preparation. As the FZD₄ structure is of higher resolution than FZD₅ (2.4 Å vs. 3.7 Å) and the only notably different area of the receptor (i.e. the extracellular part of TM6) is not resolved in the FZD₅ structure (thus offering no additional template for receptor modeling), we decided not to change our FZD₆ model based on FZD₅ structure. It should also be noted that the inactive FZD₆ model is used in this study purely for visualization purposes only, as all simulations are run with the active-like FZD₆ model as described above.

Altogether, based on the abovementioned limitations of the inactive FZD₅ structure, and the structural differences between the active and inactive receptor conformations, we feel that updating the active SMO conformation-inspired FZD₆ model, which we use as basis for our MD simulations here, to reflect the inactive FZD₅ structure would introduce weaknesses to the model rather than strengthening it.

Figure 2B. The authors showed that F6.43P mutation (in SMO) significantly reduced the BODIPY-cyclopamine binding affinity, and explained this result with a decrease in the volume of the lower pocket (high affinity binding pocket) in this mutant (page 6). To support this authors' explanation, it would be nice if they could show that this mutant does not show a difference in affinity for the upper binding site (using Δ CRD-SMO or SAG 1.3).

Inspired by the reviewer's comment, we have performed additional binding experiments using BODIPY-cyclopamine binding to the Δ CRD-SMO, which we have previously used to establish the NanoBRET-based BODIPY-cyclopamine assay (Kozielewicz et al 2020, Molecular Pharmacology). As surmised by the reviewer, the data argue that the binding of BODIPY-cyclopamine to the upper binding site is not affected. The data are presented in the **new Supplementary Figure 5**.

Page 4. The authors wrote that "FZD6 was selected for the MD simulations due to its similarity to SMO". It is not clear in what aspects the authors refer to similarity. Please clarify this.

As described in detail above, the phylogenetic analysis from Wright and Kozielewicz et al. 2019 (Nature Communications) underlines that SMO is closer to the FZD_{3/6} cluster than to the other FZD clusters (based on the full length protein sequence homology). To clarify the message, we rephrased the sentence and now it reads as follows: "The sequence of SMO is most similar to that of FZD₃ and FZD₆ (Wright and Kozielewicz et al. 2019). Of these, we selected FZD₆ for the MD simulations due to the fact that simulating FZD₆ allowed us to utilize the SMO agonist SAG1.3 in maintaining the simulated receptors in an active-like conformation in absence of the intracellular effector similar to what we have reported before (Kozielewicz et al. 2020).

Additionally, we added similarity percentages to the methods section: "The sequence identity between the template (Δ CRD-SMO) and modelled Δ CRD-FZD₆ is approximately 29% and sequence similarity approximately 48%."

Supplementary Figure 2. The FZD7 mutant appears to have lower surface expression than the wild type, but it says that the difference is not significant. What is the p-value? Which t-test was used? There is not enough information about statistical analysis of data throughout this manuscript.

The required information was added. Further, we went through the whole manuscript clarifying the statistical analysis where required.

Minor issues

Figure 1A. In the figure of class A GPCRs, aligning the structures of the C-terminal part of TM6 will show

the outward movement of the cytoplasmic part of TM6 more clearly. Also, light grey labeling is hard to see.

This suggestion indeed clarified the message of the panel. Thanks! We have used the alignment of the C terminal part of the TM6 as suggested and changed the color palette.

Page 7. The authors wrote that “The only FZD crystal structures currently available are FZD4 and FZD5”. However, the FZD5 structure is not a crystal structure but a Cryo-EM structure.

Thanks, we realized that ourselves after submission. This mistake is now corrected.

Page 17. Recent findings that SMO-tagerting small => SMO-targeting

Corrected.

Reviewer #3:

In this study, Turku et al. conduct molecular dynamics simulations and numerous BRET assays to investigate the mechanism of activation of Class F GPCRs. Class F GPCRs, which consist of the Frizzled (FZD) and Smoothed (SMO) receptors, have important physiological roles including in cell polarity, cell proliferation and embryonic development. The authors find an important role for the residue at position 6.43 (P in FZD receptors, F in SMO receptors) in FZD/SMO activation. Mutagenesis of this position has various implications including on TM6 conformation, ligand binding, G protein-coupling and beta-arrestin2 recruitment.

This study presents novel findings that shed light on the activation mechanisms of Class F GPCRs, which are currently not well understood. These results will have important consequences for drug design and our understanding of GPCR structure and activation more broadly. The manuscript encompasses a comprehensive set of experiments and the results are generally explained and presented well; the figures are clear and very visually appealing. I support the publication of this manuscript with minor revisions, as outlined below.

Thanks a lot for the positive comments.

Major comments

1. A more comprehensive sequence alignment could be generated, for example using ortholog sequences obtained from Ensembl Compara (there are at least >100, and for most receptors >200, one-to-one ortholog sequences for each Class F receptor in Ensembl). An alignment built from sequences from a larger number of species would give more robust indications of how conserved certain amino acids/positions are.

This is indeed a clever suggestion. Interestingly, we have done this in a previous paper (Wright &

Kozielewicz et al 2019, Nature Communications – Supplementary Figure 1a, b and Supplementary Data). In that study, we have done a large scale sequence alignment of over 750 mammalian and non-mammalian FZDs and SMOs, which revealed several positions that are conserved among the human paralogs, in mammals as well as across the animal kingdom. Also, the phylogenetic conservation of the residue 6.43 (F in SMO, P in FZDs) is underlined by the analysis. For the revision of the current manuscript, we have included a panel in slightly modified from the published 2019 version, with reference to the full analysis presented in the previous paper as **new Supplementary Figure 1**.

2. Many important GPCR-lipid interactions have been identified, including ones that influence activation mechanisms and TM6 conformations. Could the authors comment on their choice to use a pure POPC bilayer (and not include other lipids found in eukaryotic/mammalian membranes) in the MD simulations?

It is indeed true that membrane lipids influence activation mechanisms and active GPCR conformations. However, we decided to utilize a simplified POPC bilayer system in our simulations for following reasons: First, our computational study aimed at assessing whether amino acid 6.43 mutation changes overall SMO or FZD₆ active conformation and to be sure that we only measure the effect of this one variable, we kept the simulation systems as simplified as possible. Second, it has been reported previously that when utilizing classical all-atom molecular dynamics simulations in microsecond timescale, the changes in lipid bilayer composition (in this case POPC ± cholesterol) do not result in notable differences in the simulated GPCR conformation (Karhu et al. J. Phys. Chem. B, 123, 2609–2622, 2019; <https://doi.org/10.1021/acs.jpcc.8b10220>) – most likely due to too limited observation time provided by the method to date. When considering the biology of our studied system, cholesterol would have been the most relevant lipid to be added there as well, but unfortunately it seems that it is currently outside of the scope of the method to do that in relevant way. Thus, we decided to continue using the simplified POPC bilayer approach, which has been working robustly with the studied system before (Kozielewicz et al. 2020).

3. The text states that the MD simulations of active(-like) FZD6 and SMO ran for 1250 ns, but Fig 1e of the TM6 angles only shows 500 ns.

The total simulation time is indeed 1250 ns per each simulation system, but it consists of four independent replicas (á 500 ns, and 3x250 ns), which are plotted as separate replicas e.g. in Fig. 1e. Two of these replicas are started from the frame $t = 0$ ns (i.e. directly after the equilibration) and two from the frame $t = 250$ ns – this was done to enhance the possibilities for sampling and to increase the individuality of the simulation replicas. The sentence is now reworded as follows for clarification: “In these MD simulations, a distinct kink was observed in the TM6 of FZD₆ (angle $158.5^\circ \pm 4.5^\circ$ throughout the MD trajectory run in four independent replicas comprising 1250 ns of simulation per simulation system), whereas the TM6 remained notably straighter in SMO ($168.4^\circ \pm 4.2^\circ$; Fig. 1c–e).”

4. Are the differences in binding cavity volume between FZD/SMO WT and mutant structures statistically significant (Fig 2d)?

It is a tricky question whether molecular dynamics trajectories are suitable for such statistical comparisons mainly due to the fact that classical all-atom molecular dynamics simulations are practically unable to sample the whole conformational space of a system as large as a membrane-embedded GPCR

with the current computational power. Basically, this means that as the simulation never reaches a point when it has surely gone through all possible conformations of the system, it is rather impossible to estimate, whether the sample size provided by the current simulation is enough for conducting robust statistical tests.

Despite our thorough searching efforts, we could find only one very recent report focusing on the feasibility of analyzing MD data with a statistical test (Bruzese et al., *Sci. Rep.*, 10, 19942, 2020; <https://doi.org/10.1038/s41598-020-77072-4>). In this publication, the authors conclude that multiple-factor ANOVA could be suitable for analyzing the difference between the MD trajectory-originated measurements (in that case distances). It should be noted, however, that the simulation trajectories they analyzed were approximately six times longer per each simulation system than in our case providing larger sample size. Additionally, unlike the distance measurements, the binding site volume measurement protocol includes a manual step, where volume vertices outside the 7TM core are removed based on visual inspection. As such a manual step can also fade the statistical power, we do not feel comfortable in using any statistical test on the binding site volume data.

5. Could the authors comment on why they did not normalize the results in the Gi-BRET sensor assay for receptor surface expression (Fig 3b)?

In our experience, DVL recruitment is proportional to the amount of surface expression of the coexpressed FZD and therefore we have presented both the data with surface expression normalization (Figure 6b) and the raw data (Supplementary Figure 12). In addition, we have presented a new assay format in the revised version of the manuscript, where we analyze FZD₆ wt and P^{6.43}F mutant-induced DVL DEP recruitment as bystander BRET plotted over surface expression (cell impermeable SNAP substrate) to better define the effect of receptor mutation on DVL DEP interaction excluding receptor surface expression as a confounding factor. In the case of G protein activation and TOPflash signaling, however, the correlation of signaling and surface expression remain poorly defined and therefore a normalization cannot be justified.

6. Could the authors comment on why they used a bystander BRET assay, rather than a BRET assay like the one used to measure cPKA/NbSmo2 recruitment, to measure DVL2 recruitment? With respect to this, the inclusion of a negative control could be important to confirm that the BRET signal from the bystander BRET assay represents specific binding of DVL2 to FZD receptors.

Over the last years, we have established diverse FZD-DVL readouts including the bystander BRET assay, which reliably works in our hands (see also Wright and Kozielowicz et al 2019, *Nature Communications*; Kozielowicz et al 2020, *Nature Communications*). The assay has the advantage that the readout is restricted to plasma membrane recruitment of DVL and is not blurred by potential receptor-DVL interaction in intracellular compartments (e.g. by receptors that did not make it to the surface). The negative control that we have presented in the manuscript is SMO and SMO F^{6.43}P, which do not recruit DVL (see also Supplementary Figure 12), thereby underlining the selectivity of the readout for receptor-mediated membrane recruitment.

With regard to the cPKA/NbSMO2 sensors, we used a direct BRET mostly because the assay was recently set up by Benjamin Myers in this way (see also bioRxiv preprint: *C. D. Arveseth et al., Smoothened*

Transduces Hedgehog Signals via Activity-Dependent Sequestration of PKA Catalytic Subunits. bioRxiv, 2020.2007.2001.183079 (2020).

7. The discussion of DVL2 recruitment to FZD7 is slightly confusing: the cell surface expression-normalized results are not statistically significant (Fig 5f), so claiming that there is a “similar trend” of reduced DVL2 recruitment using the non-normalized data from the supplementary figure (Fig S7) seems misleading.

We agree, this is now reworded.

8. Could the authors comment on how including the mini-Gi protein (as done for simulations presented in the Discussion section) would influence the binding cavity volume in the active-like FZD simulations (as presented in Fig 2d)?

This is indeed very interesting question, and we conducted and added the cavity volume analysis of FZD₆-miniG_i system to **Supplementary Figure S4**. As the visual examination of the simulation trajectory already suggested, the 7TM cavity volume of the FZD₆-miniG_i follows quite well that of FZD₆ P^{6.43}F mutant as the TM2 bulge observed in the original FZD₆ wild type simulation is absent.

9. Figures are labelled with * and ** – could the authors clarify what (presumably p-values) these refer to?

This is now clarified.

Minor comments

1. More detail on the “similarity” between FZD6 and SMO, which factored into the choice to use FZD6 for the MD simulations, would be helpful (e.g. sequence similarity and if so, what percent?).

We agree that this sentence was too general. To clarify the message, we rephrased the sentence and now it reads as follows: “The sequence of SMO is most similar to that of FZD₃ and FZD₆ (Wright and Kozielowicz et al. 2019). Of these, we selected FZD₆ for the MD simulations due to the fact that simulating FZD₆ allowed us to utilize the SMO agonist SAG1.3 in maintaining the simulated receptors in an active-like conformation in absence of the intracellular effector similar to what we have reported before (Kozielowicz et al. 2020).

Additionally, we added similarity percentages to the methods section: “The sequence identity between the template (Δ CRD-SMO) and modelled Δ CRD-FZD₆ is approximately 29% and sequence similarity approximately 48%.”

2. The FZD6-SAG1.3-miniGi simulations are first mentioned in the Discussion. It would be helpful to move these results to the Results section.

FZD₆-SAG1.3-miniGi simulations are now lifted to the Results section.

Figure comments

1. Fig 1b: the top panel (appears to be a measure of sequence conservation) is not explained in the figure legend.

Thanks, this is now fixed.

2. Fig 1e: it appears as though replica 4 is starting halfway through (at 250 ns). I am not sure if this is merely a result of the data representation, but it is slightly confusing.

As briefly mentioned above, the independent replicas are run using two different starting conformations in order to enhance the sampling of the conformational space with this limited amount of simulation time (1250 ns in total) per each simulation system. We started the simulation protocol by running a single 500 ns simulation (replica 1) and then replicated it for 250 ns starting from the same starting conformation (i.e. $t = 0$ ns; replica 2). Then we run replicas 3 and 4 for 250 ns each starting from the last frame of the replica 2 (i.e. at $t = 250$ ns when plotted on the original simulation timeline). Thus, we present replicas 3 and 4 starting from time point 250 ns in the trajectory graphs.

To clarify this, we have now added these simulation protocol details to the legend of Fig. 1 and to the "Molecular dynamics (MD) simulations" section of the Materials and methods.

This study demonstrates significant and novel findings, which will advance our understanding of Class F GPCR activation and have important implications for drug design. I enjoyed reading this work and support its publication with the relevant revisions.

Alissa M Hummer

THANKS!

Reviewers' Comments:

Reviewer #1:

Remarks to the Author:

This is a revised version of the manuscript by Turku et al. about the activation mechanisms of Class F G protein coupled receptors (GPCRs) exemplified by the Smoothed receptor (SMO) and Frizzled receptors (mostly FZD6). The authors uncover a role for a conserved proline residue in TM6 of these receptors, and the helical kink induced by that proline, as well as unique differences between SMO (where this proline is not present and the corresponding residue is a Phe) and FZD's. The insights are derived computationally (through homology modeling and molecular dynamics) and then correlated with experimental assays of WT and mutant receptor function in live cells.

In the revision, the adequately addressed the suggestions made by this reviewer in relation to the computational part of the study (only a minor comment about it below). However, the concerns about the experimental parts are addressed only partially. According to the data presented in the original manuscript, the Phe(6.43)Pro mutation reduced the ability of SMO to bind to its antagonist cyclopamine, to constitutively dissociate Gai-Gbg heterotrimer, to constitutively recruit b-arrestin, and to constitutively associate with the catalytic subunit of PKA. In FZD6, the reverse Pro(6.43)Phe mutation had no effect on cyclopamine binding but appears to reduce the receptor's constitutive association with Dvl2. These findings about receptor constitutive activity were derived from plate-based BRET assays, which, as described in the major comment below, can be problematic because of mutation-induced variations in expression and localization of the components of the assay. In the revision, the authors added some limited data describing the effects of 6.43 mutations not only on constitutive but also on ligand-induced signaling by both receptors; specifically, on their response to increasing concentrations of SAG1.3; the newly added experiment (Supp Figure 11) suggests that mutation abrogates the agonist-dependent SMO/miniGai association but the reverse mutation has minimum effect on FZD6/miniGsi association. However, the FZD6 mutation is sited in the figure as P427A not P427F so it is unclear if the comparison is fair (could the authors please comment on that?) Additionally, the newly added agonist-dependent TOPFlash transcriptional reporter assay is inconclusive because neither WT nor mutant SMO or FZD6 induce the reporter activity (in other FZD's, the mutation appears to reduce the transcriptional response). So, the majority of experimental findings remain related to the constitutive activity and therefore subject to expression/localization-associated artifacts as detailed below.

Major comment

A number of BRET experiments in the paper are conducted with C-terminally NLuc-tagged receptor, including the association with Halo-tagged b-arrestin2, Venus-tagged mini-mGsi, YFP-tagged catalytic subunit of PKA, and YFP-tagged NbSmo2. For this experimental design (donor-tagged SMO), all *constitutive* association measurements are likely to suffer from artifacts and misinterpretation due to possible effects of the mutation on the expression and localization of the donor-tagged receptors. Notably, such effects may be mediated not only the surface-expressed SMO (or which ELISA-based surface expression measurements are presented in Supp Fig 6 and demonstrate substantial but inconsistent variations in mutant expression in comparison with WT SMO-NLuc) but also intracellular SMO (for which expression data is not presented; this would be easy to address by showing the overall NLuc luminescence in each experiment). In this scenario, it would be preferably to at least show the experiments in the acceptor titration mode (as is done for PKA) and not in a single-point mode (as is done for mini-Gsi and b-arrestin).

Similarly, those BRET experiments where neither the donor nor acceptor tag is on the WT or mutated receptors (such as the intermolecular between NLuc-tagged Gai and Venus-tagged Gbg, or the bystander BRET between NLuc-tagged Dvl2 and a Venus-tagged PM marker) are also subject to potential artifacts as a result of mutation-induced changes in receptor expression.

These caveats need to be clearly explained in the text and addressed where possible with (i) robust receptor construct surface and total expression measurements, (ii) acceptor titration experiments, and (iii) ligand-dependent effector association experiments.

Minor comments

* The abstract needs to be revised to include the summary of the findings from the functional experiments with mutants (currently only states that SMO and FZDs are different with much attention paid to the straightness vs kink in helix VI).

* Proline puckering dynamics (Supp Figure 3 & its legend) were added in response to this reviewer's comment from the initial submission. These are very informative but would it be possible to reduce the size of the markers on the plot or come up with an alternative visualization? As it is, the large number of magenta and pink circles simply mask the blue and violet, making the figure somewhat unclear. Same with the lower right panel of Supp Figure 4.

* Background info and schematics (similar to the first panels in all BRET figures) are needed for the HiBiT surface expression (and ligand binding) assay as well as the TOPFlash TCF/LEF transcriptional reporter assay.

Reviewer #2:

Remarks to the Author:

The revised manuscript by Turku et al. entitled "Residue 6.43 defines receptor function in Class F GPCRs", and the very detailed rebuttal letter, provided missing details and new experimental evidence supporting the proposed activation-associated conformational changes of Fzd6.

My initial concerns about the reliability of the homology model of Fzd6 based on SMO have now been resolved from the detailed explanation and new Supp.Fig.1. The authors presented new data that the P6.43F mutation interferes with DVL DEP recruitment to FZD6 (Fig. 6c) and canonical signaling by Wnt3a (Fig. 6d). The authors also performed a new binding experiment using BODIPY-cyclopamine binding to the deltaCRD-SMO (Supp.Fig.5). All issues raised in my previous comments have been satisfactorily addressed.

Reviewer #3:

Remarks to the Author:

The authors have, in my view, appropriately addressed the reviewer comments. In particular, the manuscript has been strengthened by the addition of the TOPFlash and mGsi experiments (Fig 6d and Supplementary Fig 11), which investigate the effects of the 6.43 P/F mutation on WNT signaling and mini-Gi protein recruitment, respectively. (To note though, there is a typo in Supp Fig 11: the lower right panel figure label lists the FZD6 mutation as P427*A* 6.43, while the mutation is described as P6.43*F* in the figure legend.)

It was also helpful that the authors included a clearer clarification for the homology modeling of active-like FZD6, including the specification of percent sequence identity and similarity between FZD6 and SMO – this is important for increasing confidence in the MD simulation results.

In all, I find this is a comprehensive manuscript with interesting and important findings about the mechanisms of activation in Class F GPCRs. I fully support the acceptance of this manuscript.

Alissa M Hummer

Rebuttal letter revision #2 – author's comments in green

Reviewer #1 (Remarks to the Author):

This is a revised version of the manuscript by Turku et al. about the activation mechanisms of Class F G protein coupled receptors (GPCRs) exemplified by the Smoothed receptor (SMO) and Frizzled receptors (mostly FZD6). The authors uncover a role for a conserved proline residue in TM6 of these receptors, and the helical kink induced by that proline, as well as unique differences between SMO (where this proline is not present and the corresponding residue is a Phe) and FZD's. The insights are derived computationally (through homology modeling and molecular dynamics) and then correlated with experimental assays of WT and mutant receptor function in live cells.

In the revision, the authors adequately addressed the suggestions made by this reviewer in relation to the computational part of the study (only a minor comment about it below). However, the concerns about the experimental parts are addressed only partially. According to the data presented in the original manuscript, the Phe(6.43)Pro mutation reduced the ability of SMO to bind to its antagonist cyclopamine, to constitutively dissociate Gai-Gbg heterotrimers, to constitutively recruit b-arrestin, and to constitutively associate with the catalytic subunit of PKA. In FZD6, the reverse Pro(6.43)Phe mutation had no effect on cyclopamine binding but appears to reduce the receptor's constitutive association with Dvl2. These findings about receptor constitutive activity were derived from plate-based BRET assays, which, as described in the major comment below, can be problematic because of mutation-induced variations in expression and localization of the components of the assay. In the revision, the authors added some limited data describing the effects of 6.43 mutations not only on constitutive but also on ligand-induced signaling by both receptors; specifically, on their response to increasing concentrations of SAG1.3; the newly added experiment (Supp Figure 11) suggests that mutation abrogates the agonist-dependent SMO/miniGai association but the reverse mutation has minimum effect on FZD6/miniGsi association. However, the FZD6 mutation is sited in the figure as P427A not P427F so it is unclear if the comparison is fair (could the authors please comment on that?) Additionally, the newly added agonist-dependent TOPFlash transcriptional reporter assay is inconclusive because neither WT nor mutant SMO or FZD6 induce the reporter activity (in other FZD's, the mutation appears to reduce the transcriptional response). So, the majority of experimental findings remain related to the constitutive activity and therefore subject to expression/localization-associated artifacts as detailed below.

First of all, thanks for pointing the inconsistency of the P427A vs P427F mutant of FZD₆ out. In the initial phase of the study, we analysed some P^{6.43}A and P^{6.43}F mutations in parallel. For FZD₆ in some assays these data were unintentionally carried along presenting the P to A mutant. This is now corrected with new data on the FZD₆ P^{6.43}F mutation for all instances, where the FZD₆ P^{6.43}A was mentioned in the previous version of the manuscript. The new data showing the FZD₆ P^{6.43}F mutant do by no means differ from the overall picture obtained with the FZD₆ P^{6.43}A mutant and thus, they do not affect the overall conclusions. We apologize for causing confusion.

Major comment

A number of BRET experiments in the paper are conducted with C-terminally Nluc-tagged receptor, including the association with Halo-tagged b-arrestin2, Venus-tagged mini-mGsi, YFP-tagged catalytic subunit of PKA, and YFP-tagged NbSmo2. For this experimental design (donor-tagged SMO), all *constitutive* association measurements are likely to suffer from artifacts and misinterpretation due to possible effects of the mutation on the expression and localization of the donor-tagged receptors.

Notably, such effects may be mediated not only the surface-expressed SMO (or which ELISA-based surface expression measurements are presented in Supp Fig 6 and demonstrate substantial but inconsistent variations in mutant expression in comparison with WT SMO-NLuc) but also intracellular SMO (for which expression data is not presented; this would be easy to address by showing the overall NLuc luminescence in each experiment). In this scenario, it would be preferably to at least show the experiments in the acceptor titration mode (as is done for PKA) and not in a single-point mode (as is done for mini-Gsi and b-arrestin).

Similarly, those BRET experiments where neither the donor nor acceptor tag is on the WT or mutated receptors (such as the intermolecular between NLuc-tagged Gai and Venus-tagged Gbg, or the bystander BRET between NLuc-tagged Dvl2 and a Venus-tagged PM marker) are also subject to potential artifacts as a result of mutation-induced changes in receptor expression.

These caveats need to be clearly explained in the text and addressed where possible with (i) robust receptor construct surface and total expression measurements, (ii) acceptor titration experiments, and (iii) ligand-dependent effector association experiments.

Thanks a lot for your constructive criticism.

We completely agree with the reviewer that receptor surface expression is an important factor in the interpretation of the presented data. Therefore, we have made an effort to provide a comparative surface expression analysis for all constructs (wt and mutants) used in this study.

In order to address the recommendations by this reviewer, we have:

1. We introduced full titration BRET curves for FZD₇ and SMO in Fig. 3. We chose these FZDs, because mutation of the respective receptors showed a difference in the constitutive interaction with b-arrestin. The other receptors did not show substantial constitutive b-arrestin recruitment. Thus, we did not further investigate those with full titration curves.
2. We have added a paragraph in the discussion to underline and clarify the limitations regarding the surface expression vs functional data.

Furthermore, we would like to explain and discuss the relationship between surface expression data and receptor construct performance in different assays underlining that our presentation of surface expression and effect suffices to basically exclude these potential artefacts. In fact, the data included in the manuscript argue against the reviewers concern that different surface expression levels of the receptors affect our conclusions. For example: Mutant SNAP-FZD₄ P^{6.43}F presents with slightly lower surface expression compared to wt (Suppl Fig. 6) but shows comparable DVL2 recruitment for wt and mutant (Suppl Fig 11; Fig. 6b) and decreased ability of the mutant to mediate a WNT-3A-induced TOPflash signal (Fig. 6d). On the contrary, SNAP-FZD₆ P^{6.43}F shows similar surface expression as wt (Suppl Fig. 6), but is substantially impaired in DVL2 recruitment (Fig. 6b). Furthermore, we elaborate on this finding showing that the mutant FZD₄ is also impaired in DEP recruitment using an assay that directly takes surface expression into consideration (Fig. 6c). Similarly, NLuc tagged FZD₇ and SMO, which both show variable surface expression (not statistically different from wt! Suppl Fig. 6) behave differently in the BRET-based G protein association assay, where FZD₇-G protein remains unchanged in the mutant, whereas SMO-G protein coupling is completely abrogated. If changes in surface expression would be accountable for the changes in receptor function they would occur to similar degree and direction in all assays.

We do not agree with the reviewer that inclusion of the total receptor expression e.g. by reporting the Nluc counts would provide useful information, because this readout includes also e.g. immature FZDs in the ER skewing the picture the other way.

Furthermore, we understand the reviewer's request for assessment of ligand-induced vs constitutive activity because this approach might reduce the caveat of surface expression within a given data set. However, we would like to clearly underline that assessing ligand-induced changes introduces an additional variability, which is very difficult to control for in a comparative set up. Among FZDs and especially SMO in comparison to FZDs, ligands show diverse selectivity patterns, affinities and even potentially functional selectivity. This complexity was the reason to focus on constitutive activity rather than ligand-induced activity. Thus, we feel at this point that the experiments that were introduced in the first revision employing agonists are sufficient to underline potential differences.

Minor comments

* The abstract needs to be revised to include the summary of the findings from the functional experiments with mutants (currently only states that SMO and FZDs are different with much attention paid to the straightness vs kink in helix VI).

We have reworded parts of the abstract to address the reviewer's request. Thanks a lot for this constructive comment, which we feel was particularly useful.

* Proline puckering dynamics (Supp Figure 3 & its legend) were added in response to this reviewer's comment from the initial submission. These are very informative but would it be possible to reduce the size of the markers on the plot or come up with an alternative visualization? As it is, the large number of magenta and pink circles simply mask the blue and violet, making the figure somewhat unclear. Same with the lower right panel of Supp Figure 4.

We changed the symbol size in the graphs of Fig. S3 and Fig. S4 to increase readability.

* Background info and schematics (similar to the first panels in all BRET figures) are needed for the HiBiT surface expression (and ligand binding) assay as well as the TOPFlash TCF/LEF transcriptional reporter assay.

In order to address the reviewer's comment, we have added schemes to Fig. S5 (HiBiT binding), Fig. S6 (HiBiT surface expression) and Fig. 6e (TOPflash transcriptional reporter assay).

Reviewer #2 (Remarks to the Author):

The revised manuscript by Turku et al. entitled "Residue 6.43 defines receptor function in Class F GPCRs", and the very detailed rebuttal letter, provided missing details and new experimental evidence supporting the proposed activation-associated conformational changes of Fzd6.

My initial concerns about the reliability of the homology model of Fzd6 based on SMO have now been resolved from the detailed explanation and new Supp.Fig.1. The authors presented new data that the P6.43F mutation interferes with DVL DEP recruitment to FZD6 (Fig. 6c) and canonical signaling by Wnt3a (Fig. 6d). The authors also performed a new binding experiment using BODIPY-cyclopamine binding to the deltaCRD-SMO (Supp.Fig.5). All issues raised in my previous comments have been satisfactorily addressed.

We are glad that the revised manuscript addressed this reviewer's comments sufficiently.

Reviewer #3 (Remarks to the Author):

The authors have, in my view, appropriately addressed the reviewer comments. In particular, the manuscript has been strengthened by the addition of the TOPFlash and mGsi experiments (Fig 6d and Supplementary Fig 11), which investigate the effects of the 6.43 P/F mutation on WNT signaling and mini-Gi protein recruitment, respectively. (To note though, there is a typo in Supp Fig 11: the lower right panel figure label lists the FZD6 mutation as P427*A* 6.43, while the mutation is described as P6.43*F* in the figure legend.)

First of all, thanks for pointing the inconsistency of the P427A vs P427F mutant of FZD₆ out, which was also caught by reviewer #1. In the initial phase of the study, we analysed some P^{6.43}A and P^{6.43}F mutations in parallel. For FZD₆ in some assays these data were unintentionally carried along presenting the P to A mutant. This is now corrected with new data on the FZD₆ P^{6.43}F mutation for all instances, where the FZD₆ P^{6.43}A was mentioned in the previous version of the manuscript. The new data showing the FZD₆ P^{6.43}F mutant do by no means differ from the overall picture obtained with the FZD₆ P^{6.43}A mutant and thus, they do not affect the overall conclusions. We apologize for causing confusion.

It was also helpful that the authors included a clearer clarification for the homology modeling of active-like FZD6, including the specification of percent sequence identity and similarity between FZD6 and SMO – this is important for increasing confidence in the MD simulation results.

In all, I find this is a comprehensive manuscript with interesting and important findings about the mechanisms of activation in Class F GPCRs. I fully support the acceptance of this manuscript.

Alissa M Hummer

We thank this reviewer for the positive evaluation.

Reviewers' Comments:

Reviewer #1:

Remarks to the Author:

This is the second revision of the manuscript by Turku et al. about the activation mechanisms of Class F G protein coupled receptors (GPCRs) exemplified by the Smoothed receptor (SMO) and Frizzled receptors (mostly FZD6). In the revision, the authors partially addressed my comments to the experimental section of the manuscript; some by writing and some by additional experiments. It is especially encouraging (with the caveat below) to see the acceptor titration curves in Fig. 4, referred to as Fig. 3 in the rebuttal: although the data is very noisy and hardly follows the expected sigmoidal pattern, it demonstrates a significant *increase*, Bmax-wise, in constitutive b-arrestin association of SMO upon the introduction of F(6.34)P mutation, and a slight and, by visual inspection, insignificant *increase* in such association for FZD7 upon the introduction of the reverse P(6.43)F mutation. Upon introduction of the acceptor titration curves in the revision, the authors removed these four constructs from the single-acceptor-donor ratio panel in Fig. 4b. What I don't understand though, is that in the previous revision, when the single-ratio data *was* presented Fig. 4b, it demonstrated a slight but significant *increase* for the FZD7 P(6.43)F mutant (no longer observable, Fig. 4c in the revision) and a very significant *decrease* for the SMO F(6.34)P mutant (now appears to be an *increase*, Fig. 4d in the revision). Why the opposite result in the new revision? Furthermore, the authors used the extremely noisy curves in Figs. 4c and 4d to derive the *affinity* (!) of the association between the receptors and b-arrestin; this type of analysis suggested a *decrease* in SMO F(6.43)P association affinity compared to WT (Kd going down from 763.1 ± 227.9 to 277.7 ± 81.0 , consistent with the interpretation presented in revision 2 but contradicting the actual data in revision 2), with the reverse FZD7 mutation causing no significant effect.

This is all very confusing even to me as a reviewer, let alone to the future reader. I do understand and appreciate the differences in BRETmax and BRET50, but not the attempts to derive the latter from the data in Figs 4c and 4d, or the discrepancies between the original data and the data in the revision. To be honest, my impression is that none of the b-arrestin BRET data can be confidently be claimed to be evidence of an increase or a decrease of association upon the introduction of the mutation: it is too noisy, affected by mutation-induced variations of receptor expression, and overall, not to the standard of Nature Communications. I am not even convinced that the data suggests association at all. I suggest that the authors either provide cleaner data or remove claims of mutation effects on b-arrestin recruitment. Although the acceptor titration curves are not presented for G protein association data, it appears to be more believable, by deltaBRET if not by anything else; the cPKA and Nb2 acceptor titration data is much more interpretable too, and so is DEP recruitment. So, hopefully, the removal of b-arrestin claims will not influence the overall conclusions of the manuscript or the prospects of its publishing in Nature Communications (I still find the subject of manuscript and the breadth of approaches to address it to very interesting and worthy of a high-profile journal like NComms).

Rebuttal letter:

See the author's reply in green

Reviewer #1 (Remarks to the Author):

This is the second revision of the manuscript by Turku et al. about the activation mechanisms of Class F G protein coupled receptors (GPCRs) exemplified by the Smoothed receptor (SMO) and Frizzled receptors (mostly FZD6). In the revision, the authors partially addressed my comments to the experimental section of the manuscript; some by writing and some by additional experiments. It is especially encouraging (with the caveat below) to see the acceptor titration curves in Fig. 4, referred to as Fig. 3 in the rebuttal: although the data is very noisy and hardly follows the expected sigmoidal pattern, it demonstrates a significant **increase**, Bmax-wise, in constitutive b-arrestin association of SMO upon the introduction of F(6.34)P mutation, and a slight and, by visual inspection, insignificant **increase** in such association for FZD7 upon the introduction of the reverse P(6.43)F mutation. Upon introduction of the acceptor titration curves in the revision, the authors removed these

four constructs from the single-acceptor-donor ratio panel in Fig. 4b. What I don't understand though, is that in the previous revision, when the single-ratio data **was** presented Fig. 4b, it demonstrated a slight but significant **increase** for the FZD7 P(6.43)F mutant (no longer observable, Fig. 4c in the revision) and a very significant **decrease** for the SMO F(6.34)P mutant (now appears to be an **increase**, Fig. 4d in the revision). Why the opposite result in the new revision? Furthermore, the authors used the extremely noisy curves in Figs. 4c and 4d to derive the **affinity** (!) of the association between the receptors and b-arrestin; this type of analysis suggested a **decrease** in SMO F(6.43)P association affinity compared to WT (Kd going down from 763.1 ± 227.9 to 277.7 ± 81.0 , consistent with the interpretation presented in revision 2 but contradicting the actual data in revision 2), with the reverse FZD7 mutation causing no significant effect.

This is all very confusing even to me as a reviewer, let alone to the future reader. I do understand and appreciate the differences in BRETmax and BRET50, but not the attempts to derive the latter from the data in Figs 4c and 4d, or the discrepancies between the original data and the data in the revision. To be honest, my impression is that none of the b-arrestin BRET data can be confidently be claimed to be evidence of an increase or a decrease of association upon the introduction of the mutation: it is too noisy, affected by mutation-induced variations of receptor expression, and overall, not to the standard of Nature Communications. I am not even convinced that the data suggests association at all. I suggest that the authors either provide cleaner data or remove claims of mutation effects on b-arrestin recruitment. Although the acceptor titration curves are not presented for G protein association data, it appears to be more believable, by deltaBRET if not by anything else; the

cPKA and Nb2 acceptor titration data is much more interpretable too, and so is DEP recruitment. So, hopefully, the removal of b-arrestin claims will not influence the overall conclusions of the manuscript or the prospects of its publishing in Nature Communications (I still find the subject of manuscript and the

breadth of approaches to address it to very interesting and worthy of a high-profile journal like NComms).

We thank this reviewer for clarity in the comments and we agree that

1. Lifting out the arrestin data in Fig. 4 increases clarity of the manuscript.
2. The removal of the arrestin data does not affect the overall conclusions of the manuscript.

This comment was very constructive for the quality of the manuscript.

Thus, we removed Figure 4 from the manuscript and adapted the manuscript to this change.

Reviewers' Comments:

Reviewer #1:

Remarks to the Author:

I'd like to thank the authors for addressing my remaining concern and agreeing to exclude the inconclusive b-arrestin data from the manuscript. I believe that in its present form, the manuscript is suitable for publication in NComms.